# Endogenous hydrogen peroxide positively regulates secretion of a gut-derived peptide in neuroendocrine potentiation of the oxidative stress response in *Caenorhabditis elegans*

Qi Jia[1,2], Drew Young[3,4], Qixin Zhang[2,4], Derek Sieburth[4,5]*

[1]Development, Stem Cells and Regenerative Medicine PhD program, Keck School of Medicine, University of Southern California, Los Angeles, United States; [2]Neuromedicine Graduate Program, University of Southern California, Los Angeles, United States; [3]Neuroscience Graduate Program, University of Southern California, Los Angeles, United States; [4]Zilkha Neurogenetic Institute, University of Southern California, Los Angeles, United States; [5]Department of Physiology and Neuroscience, Keck School of Medicine, University of Southern California, Los Angeles, United States

*For correspondence:
sieburth@usc.edu

Competing interest: The authors declare that no competing interests exist.

## eLife Assessment

This study presents **convincing** evidence of the role of an intestine-released neuropeptide, FLP-2, in the oxidative stress response of *C. elegans*, as well as for the neural circuit pathway that regulates its release in response to sensing reactive oxygen species (i.e., H2O2). These **valuable** results advance the understanding of gut-brain signaling and the neural circuit basis of behavioral responses to stress.

**Abstract** The gut-brain axis mediates bidirectional signaling between the intestine and the nervous system and is critical for organism-wide homeostasis. Here, we report the identification of a peptidergic endocrine circuit in which bidirectional signaling between neurons and the intestine potentiates the activation of the antioxidant response in *Caenorhabditis elegans* in the intestine. We identify an FMRF-amide-like peptide, FLP-2, whose release from the intestine is necessary and sufficient to activate the intestinal oxidative stress response by promoting the release of the antioxidant FLP-1 neuropeptide from neurons. FLP-2 secretion from the intestine is positively regulated by endogenous hydrogen peroxide ($H_2O_2$) produced in the mitochondrial matrix by *sod-3*/superoxide dismutase, and is negatively regulated by *prdx-2*/peroxiredoxin, which depletes $H_2O_2$ in both the mitochondria and cytosol. $H_2O_2$ promotes FLP-2 secretion through the DAG and calcium-dependent protein kinase C family member *pkc-2* and by the SNAP25 family member *aex-4* in the intestine. Together, our data demonstrate a role for intestinal $H_2O_2$ in promoting inter-tissue antioxidant signaling through regulated neuropeptide-like protein exocytosis in a gut-brain axis to activate the oxidative stress response.

## Introduction

The gut-brain axis is critical for communication between the intestine and the nervous system to regulate behavior and maintain homeostasis, and altered gut-brain signaling is associated with neurodegeneration, obesity, and tumor proliferation (*Carabotti et al., 2015*; *Grenham et al., 2011*; *Mayer*

*et al., 2022*; *Mehrian-Shai et al., 2019*; *Vitali et al., 2022*). Over the last decade the importance of peptides that function as signals in gut-brain signaling has gained recognition. Numerous gut peptides are distributed throughout the gastrointestinal (GI) tract with regional specificity (*Haber et al., 2017*), and gut-secreted peptides can modulate neurocircuits that regulate feeding behavior and glucose metabolism (*Batterham and Bloom, 2003*; *Han et al., 2018*; *Song et al., 2019*), inflammatory responses against pathogenic bacteria (*Campos-Salinas et al., 2014*; *Yu et al., 2021*), and satiety (*Batterham et al., 2002*; *Chelikani et al., 2005*; *Gibbs et al., 1973*; *Lutz et al., 1995*; *Lutz et al., 1994*; *West et al., 1984*). A gut-released peptide suppresses arousal through dopaminergic neurons during sleep in *Drosophila* (*Titos et al., 2023*). In *Caenorhabditis elegans*, gut-derived peptides regulate rhythmic behavior and behavioral responses to pathogenic bacteria (*Lee and Mylonakis, 2017*; *Singh and Aballay, 2019*; *Wang et al., 2013*). Conversely, peptides released from the nervous system regulate many aspects of intestinal function including gut mobility, inflammation, and immune defense (*Browning and Travagli, 2014*; *Furness et al., 2014*; *Lai et al., 2017*). In *C. elegans*, the secretion of peptides from various neurons regulates the mitochondrial unfolded protein response (UPR$^{mt}$), the heat shock response, and the antioxidant response in the intestine (*Jia and Sieburth, 2021*; *Maman et al., 2013*; *Prahlad et al., 2008*; *Shao et al., 2016*). In spite of the many roles of peptides in the gut-brain axis, the mechanisms underlying the regulation of intestinal peptide secretion and signaling remain to be fully defined.

Hydrogen peroxide ($H_2O_2$) is emerging as an important signaling molecule that regulates intracellular signaling pathways by modifying specific reactive residues on target proteins. For example, $H_2O_2$-regulated phosphorylation of inhibitor of nuclear factor κB (NF-κB) kinase leads to the activation of NF-κB during development, inflammation, and immune responses (*Kamata et al., 2002*; *Oliveira-Marques et al., 2009*; *Takada et al., 2003*). In addition, $H_2O_2$-induced tyrosine and cysteine modifications contribute to redox regulation of c-Jun N-terminal kinase 2 (JNK2), Src family kinase, extracellular signal-regulated kinases 1 and 2 (ERK1/2), protein kinase C (PKC), and other protein kinases (*Kemble and Sun, 2009*; *Konishi et al., 1997*; *Lee et al., 2003*; *Nelson et al., 2018*). $H_2O_2$ signaling has been implicated in regulating neurotransmission and transmitter secretion. $H_2O_2$ at low concentration increases neurotransmission at neuromuscular junctions without influencing lipid oxidation (*Giniatullin and Giniatullin, 2003*; *Giniatullin et al., 2019*; *Shakirzyanova et al., 2009*), and enhanced endogenous $H_2O_2$ generation regulates dopamine release (*Avshalumov et al., 2005*; *Avshalumov and Rice, 2003*; *Bao et al., 2005*; *Chen et al., 2001a*; *Chen et al., 2002*). Acute $H_2O_2$ treatment increases exocytosis of ATP-containing vesicles in astrocytes (*Li et al., 2019*). Finally, mitochondrially derived $H_2O_2$ regulates neuropeptide release from neurons in *C. elegans* (*Jia and Sieburth, 2021*). Cellular $H_2O_2$ levels are tightly controlled through the regulation of its production from superoxide by superoxide dismutases (SODs) and cytoplasmic oxidases (*Fridovich, 1995*; *Fridovich, 1997*; *Messner and Imlay, 2002*; *Zelko et al., 2002*), and through its degradation by catalases, peroxidases, and peroxiredoxins (*Chance et al., 1979*; *Marinho et al., 2014*). In the intestine, endogenously produced $H_2O_2$ plays important roles as an antibacterial agent in the lumen, and in activating the ER unfolded protein response (UPR$^{ER}$) through protein sulfenylation (*Botteaux et al., 2009*; *Corcionivoschi et al., 2012*; *Hourihan et al., 2016*; *Miller et al., 2020*).

Here, we demonstrate a role for endogenous $H_2O_2$ signaling in the intestine in regulating the release of the intestinal FMRF-amide-like peptide, FLP-2, to modulate a neurocircuit that activates the antioxidant response in the intestine in *C. elegans*. Intestinal FLP-2 signaling functions by potentiating the release of the antioxidant neuropeptide-like protein FLP-1 from AIY interneurons, which in turn activates the antioxidant response in the intestine. FLP-2 secretion from the intestine is rapidly and positively regulated by $H_2O_2$, whose levels are positively regulated by superoxide dismutases in the mitochondrial matrix and cytosol, and negatively regulated by the peroxiredoxin-thioredoxin system in the cytosol. Intestinal FLP-2 release is mediated by *aex-4*/SNAP25-dependent exocytosis of dense core vesicles (DCVs) and $H_2O_2$-induced FLP-2 secretion is dependent upon the production of intestinal diacylglycerol (DAG) and on *pkc-2*/PKCα/β kinase activity.

## Results

### Neuronal FLP-1 secretion is regulated by neuropeptide signaling from the intestine

We previously showed that 10 min treatment with the mitochondrial toxin juglone leads to a rapid, reversible, and specific increase in FLP-1 secretion from AIY, as measured by a twofold increase in coelomocyte fluorescence in animals expressing FLP-1::Venus fusion proteins in AIY (*Figure 1A and B*; *Jia and Sieburth, 2021*). Coelomocytes take up secreted neuropeptides by bulk endocytosis (*Fares and Greenwald, 2001*) and the fluorescence intensity of Venus in their endocytic vacuoles is used as a measure of regulated neuropeptide secretion efficacy (*Ailion et al., 2014*; *Ch'ng et al., 2008*; *Sieburth et al., 2007*). To determine the role of the intestine in regulating FLP-1 secretion, we first examined *aex-5* mutants. *aex-5* encodes an intestinal subtilisin/kexin type 5 prohormone convertase that functions to proteolytically process peptide precursors into mature peptides in DCVs (*Edwards et al., 2019*; *Thacker and Rose, 2000*), and *aex-5* mutants are defective in peptide signaling from the intestine (*Mahoney et al., 2008*). We found that *aex-5* mutants expressing FLP-1::Venus in AIY exhibited no significant difference in coelomocyte fluorescence compared to wild-type controls in the absence of juglone (*Figure 1B*). However, coelomocyte fluorescence did not significantly increase in *aex-5* mutants treated with juglone. Expression of *aex-5* cDNA selectively in the intestine (under the *ges-1* promoter) fully restored normal responses to juglone to *aex-5* mutants, whereas *aex-5* cDNA expression in the nervous system (under the *rab-3* promoter) failed to rescue (*Figure 1B*). Thus, peptide processing in intestinal DCVs is necessary for juglone-induced FLP-1 secretion from AIY.

Next, we examined a number of mutants with impaired SNARE-mediated vesicle release in the intestine including *aex-1*/UNC13, *aex-3*/MADD, *aex-4*/SNAP25b, and *aex-6*/Rab27 (*Figure 2C*; *Iwasaki et al., 1997*; *Mahoney et al., 2006*; *Thacker and Rose, 2000*; *Thomas, 1990*; *Wang et al., 2013*), and they each exhibited no increases in FLP-1 secretion following juglone treatment above levels observed in untreated controls (*Figure 1—figure supplement 1A*). NLP-40 is a neuropeptide-like protein whose release from the intestine is presumed to be controlled by *aex-1*, *aex-3*, *aex-4*, and *aex-6* (*Lin-Moore et al., 2021*; *Mahoney et al., 2008*; *Shi et al., 2022*; *Wang et al., 2013*). Null mutants in *nlp-40* or its receptor, *aex-2* (*Wang et al., 2013*), exhibited normal juglone-induced FLP-1 secretion (*Figure 1—figure supplement 1B*). These results establish a gut-to-neuron signaling pathway that regulates FLP-1 secretion from AIY that is likely to be controlled by peptidergic signaling distinct from NLP-40.

### FLP-2 signaling from the intestine potentiates neuronal FLP-1 secretion and the oxidative stress response

*flp-1* protects animals from the toxic effects of juglone (*Jia and Sieburth, 2021*). We reasoned that the intestinal signal that regulates FLP-1 secretion should also protect animals from juglone-induced toxicity. We identified the FMRF-amide neuropeptide-like protein, *flp-2*, in an RNA interference (RNAi) screen for neuropeptides that confer hypersensitivity to juglone toxicity upon knockdown (*Jia and Sieburth, 2021*). *flp-2* signaling has been implicated in regulating lifespan, reproductive development, locomotion during lethargus, and the mitochondrial unfolded protein response (UPR^mt) (*Chai et al., 2022*; *Chen et al., 2016*; *Kageyama et al., 2022*; *Shao et al., 2016*). Putative *flp-2(ok3351)* null mutants, which eliminate most of the *flp-2* coding region, are superficially as healthy as wild-type animals, but they exhibited significantly reduced survival in the presence of juglone compared to wild-type controls (*Figure 1C*). The reduced survival rate of *flp-2* mutants was similar to that of *flp-1* mutants, and *flp-1; flp-2* double mutants exhibited survival rates that were not more severe than those of single mutants (*Figure 1C*), suggesting that *flp-1* and *flp-2* may function in a common genetic pathway.

To determine whether *flp-2* signaling regulates FLP-1 secretion from AIY, we examined FLP-1::Venus secretion. *flp-2* mutants exhibited normal levels of FLP-1 secretion in the absence of stress, but FLP-1 secretion failed to significantly increase following juglone treatment of *flp-2* mutants (*Figure 1D*). *flp-2* is expressed in a subset of neurons as well as the intestine (*Chai et al., 2022*), and *flp-2* functions from the nervous system for its roles in development and the UPR^mt (*Chai et al., 2022*; *Chen et al., 2016*; *Kageyama et al., 2022*; *Shao et al., 2016*). Expressing a *flp-2* genomic DNA, fragment (containing both the *flp-2a* and *flp-2b* isoforms that arise by alternative splicing), specifically in the nervous system

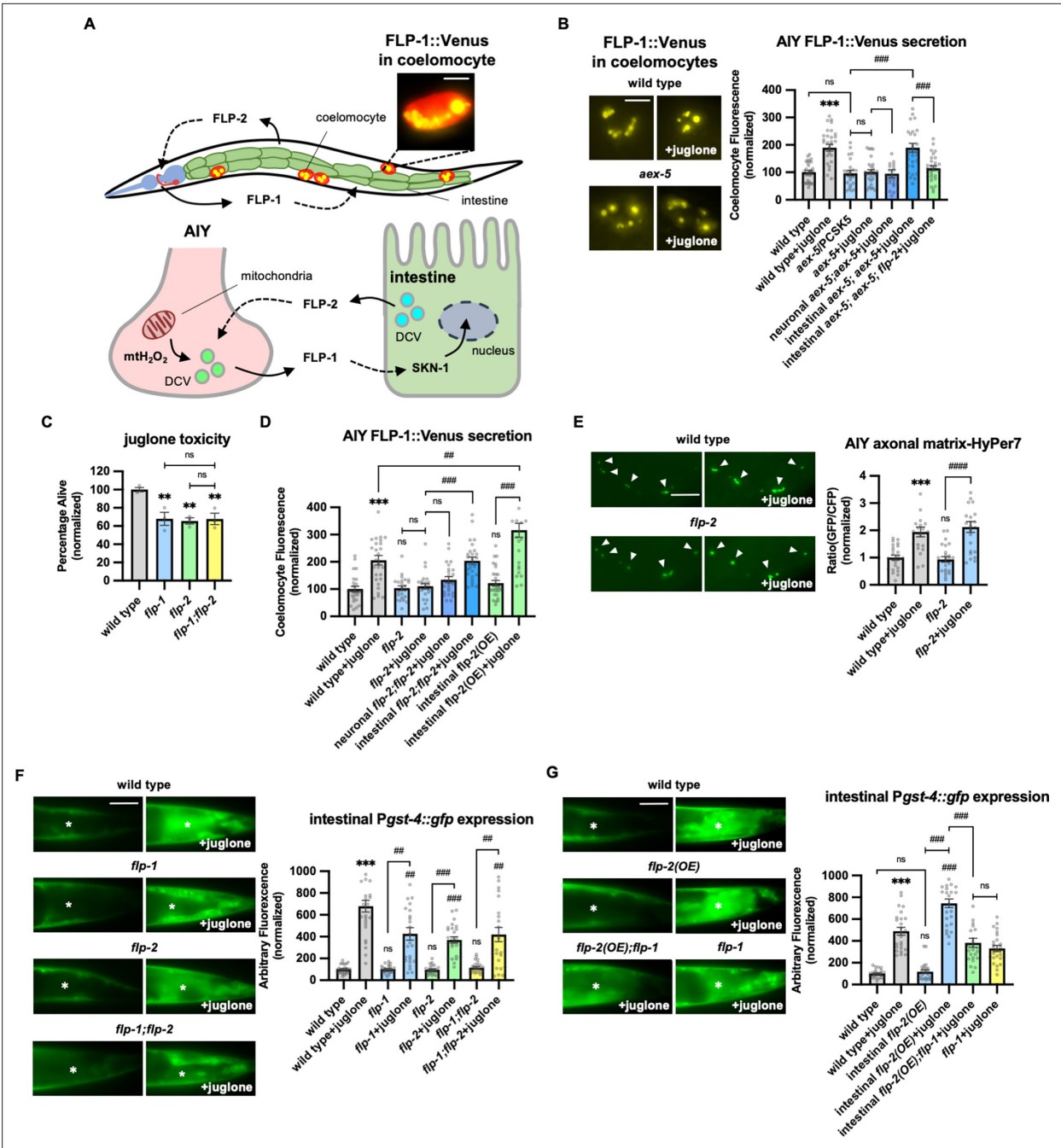

**Figure 1.** Peptidergic gut-to-neuron FLP-2 signaling potentiates the oxidative stress response. (**A**) (Top) Schematic showing the positions of AIY, intestine, and coelomocytes of transgenic animals co-expressing FLP-1::Venus in the intestine and mCherry in coelomocytes. Representative image of the posterior coelomocyte that has taken up Venus into the endocytic compartment. Scale bar: 5 μM. (Bottom) Schematic showing FLP-1 and FLP-2 peptides as inter-tissue signals in gut-intestine regulation of the antioxidant response. (**B**) Representative images and quantification of average coelomocyte fluorescence of the indicated mutants expressing FLP-1::Venus fusion proteins in AIY following M9 or 300 μM juglone treatment for 10 min. Neuronal *aex-5* denotes expression of *aex-5* cDNA under the *rab-3* promoter; intestinal *aex-5* denotes expression of *aex-5* cDNA under the *ges-1* promoter. Unlined *** denotes statistical significance compared to 'wild type'. n=30, 30, 24, 30, 26, 30, 30 independent animals. Scale bar: 5 μM. (**C**) Average percentage of surviving young adult animals of the indicated genotypes after 16 hr recovery following 4 hr juglone treatment. Unlined ** denotes statistical significance compared to 'wild type'. n=213, 156, 189, 195 independent biological samples over three independent experiments. (**D**) Quantification of average coelomocyte fluorescence of the indicated mutants expressing FLP-1::Venus fusion proteins in AIY following M9 or 300 μM juglone treatment for 10 min. Neuronal *flp-2* denotes expression of *flp-2* gDNA under the *rab-3* promoter; intestinal *flp-2* denotes expression of *flp-2* gDNA under the *ges-1* promoter; intestinal *flp-2(OE)* denotes expression of *flp-2* gDNA under the *ges-1* promoter in wild-type animals. Unlined *** and ns denote statistical significance compared to 'wild type'. n=20, 20, 25, 20, 20, 20, 25, 22 independent animals. (**E**) Representative images and

*Figure 1 continued on next page*

*Figure 1 continued*

quantification of fluorescence of mitochondrial matrix-targeted HyPer7 in the axon of AIY following M9 or 300 µM juglone treatment for 10 min. Arrowheads denote puncta marked by mito::HyPer7 fusion proteins (excitation: 500 and 400 nm; emission: 520 nm). Ratio of images taken with 500 nM (GFP) and 400 nM (CFP) for excitation was used to measure $H_2O_2$ levels. Unlined *** and ns denote statistical significance compared to 'wild type'. n=24, 22, 25, 24 independent animals. Scale bar: 10 µM. (**F**) Representative images and quantification of average fluorescence in the posterior intestine of transgenic animals expressing P*gst-4::gfp* after 1h M9 or juglone exposure and 3 hr recovery. Asterisks mark the intestinal region used for quantification. P*gst-4::gfp* expression in the body wall muscles, which appears as fluorescence on the edge animals in some images, was not quantified. Unlined *** and ns denote statistical significance compared to 'wild type'; unlined ## and ### denote statistical significance compared to 'wild type+juglone'. n=25, 26, 25, 25, 25, 25, 25, 25 independent animals. Scale bar: 10 µM. (**G**) Representative images and quantification of average fluorescence in the posterior region of transgenic animals expressing P*gst-4::gfp* after 1 hr M9 or juglone exposure and 3 hr recovery. Asterisks mark the intestinal region for quantification. P*gst-4::gfp* expression in the body wall muscles, which appears as fluorescence on the edge animals in some images, was not quantified. Unlined *** denotes statistical significance compared to 'wild type'; unlined ### denotes statistical significance compared to 'wild type+juglone'. n=23, 25, 25, 26, 24, 25 independent animals. Scale bar: 10 µM. (**B–G**) Data are mean values ± s.e.m. normalized to wild-type controls. ns, not significant, ** and ## p<0.01, *** and ### p<0.001 by Brown-Forsythe and Welch ANOVA with Dunnett's T3 multiple comparisons test.

The online version of this article includes the following source data and figure supplement(s) for figure 1:

**Source data 1.** Raw data used for plotting the figures.

**Figure supplement 1.** The effect of intestinal dense core vesicle (DCV) secretion mutations on FLP-1 release from AIY.

**Figure supplement 1—source data 1.** Raw data used for plotting the figures.

failed to rescue the FLP-1::Venus defects of *flp-2* mutants, whereas expressing *flp-2* selectively in the intestine fully restored juglone-induced FLP-1::Venus secretion to *flp-2* mutants (***Figure 1D***). These results indicate that *flp-2* signaling is dispensable for FLP-1 secretion from AIY under normal conditions, but that *flp-2* originating from the intestine is necessary to increase FLP-1 secretion during oxidative stress.

To address how *flp-2* signaling regulates FLP-1 secretion from AIY, we examined $H_2O_2$ levels in AIY using a mitochondrially targeted pH-stable $H_2O_2$ sensor HyPer7 (mito-HyPer7, ***Pak et al., 2020***). Mito-HyPer7 adopted a punctate pattern of fluorescence in AIY axons, and the average fluorescence intensity of axonal mito-HyPer7 puncta increased about twofold following 10 min juglone treatment (***Figure 1E***), in agreement with our previous studies using HyPer (***Jia and Sieburth, 2021***), confirming that juglone rapidly increases mitochondrial AIY $H_2O_2$ levels. *flp-2* mutations had no significant effects on the localization or the average intensity of mito-HyPer7 puncta in AIY axons either in the absence of juglone or in the presence of juglone (***Figure 1E***), suggesting that *flp-2* signaling promotes FLP-1 secretion by a mechanism that does not increase $H_2O_2$ levels in AIY. Consistent with this, intestinal overexpression of *flp-2* had no effect on FLP-1::Venus secretion in the absence of juglone, but significantly enhanced the ability of juglone to increase FLP-1 secretion (***Figure 1D***). We conclude that both elevated mitochondrial $H_2O_2$ levels and intact *flp-2* signaling from the intestine are necessary to increase FLP-1 secretion from AIY.

Previously we showed that FLP-1 signaling from AIY positively regulates the activation of the antioxidant transcription factor SKN-1/Nrf2 in the intestine. Specifically, *flp-1* mutations impair the juglone-induced expression of the SKN-1 reporter transgene P*gst-4::gfp* (***Figure 1F***; ***Jia and Sieburth, 2021***). We found that mutations in *flp-2* caused a similar reduction in juglone-induced P*gst-4::gfp* expression as *flp-1* mutants, and that *flp-1; flp-2* double mutants exhibited similar impairments in juglone-induced P*gst-4::gfp* expression as *flp-1* or *flp-2* single mutants (***Figure 1F***). Conversely, overexpression of *flp-2* selectively in the intestine elevated juglone-induced P*gst-4::gfp* expression, without altering baseline P*gst-4::gfp* expression, and the elevated P*gst-4::gfp* expression in juglone-treated animals overexpressing *flp-2* was entirely dependent upon *flp-1* (***Figure 1G***). It's noteworthy that overexpressing *flp-2* in the intestine did not enhance FLP-1::Venus release or P*gst-4::gfp* expression in the absence of stress, indicating that the FLP-1 mediated antioxidant pathway by *flp-2* is stress-activated. Together this data indicates that *flp-2* signaling originating in the intestine positively regulates the stress-induced secretion of FLP-1 from AIY, as well as the subsequent activation of antioxidant response genes in the intestine. We propose that FLP-1 and FLP-2 define a bidirectional gut-neuron signaling axis, whereby during periods of oxidative stress, FLP-2 released from the intestine positively regulates FLP-1 secretion from AIY, and FLP-1, in turn, potentiates the antioxidant response in the intestine (***Figure 1A***).

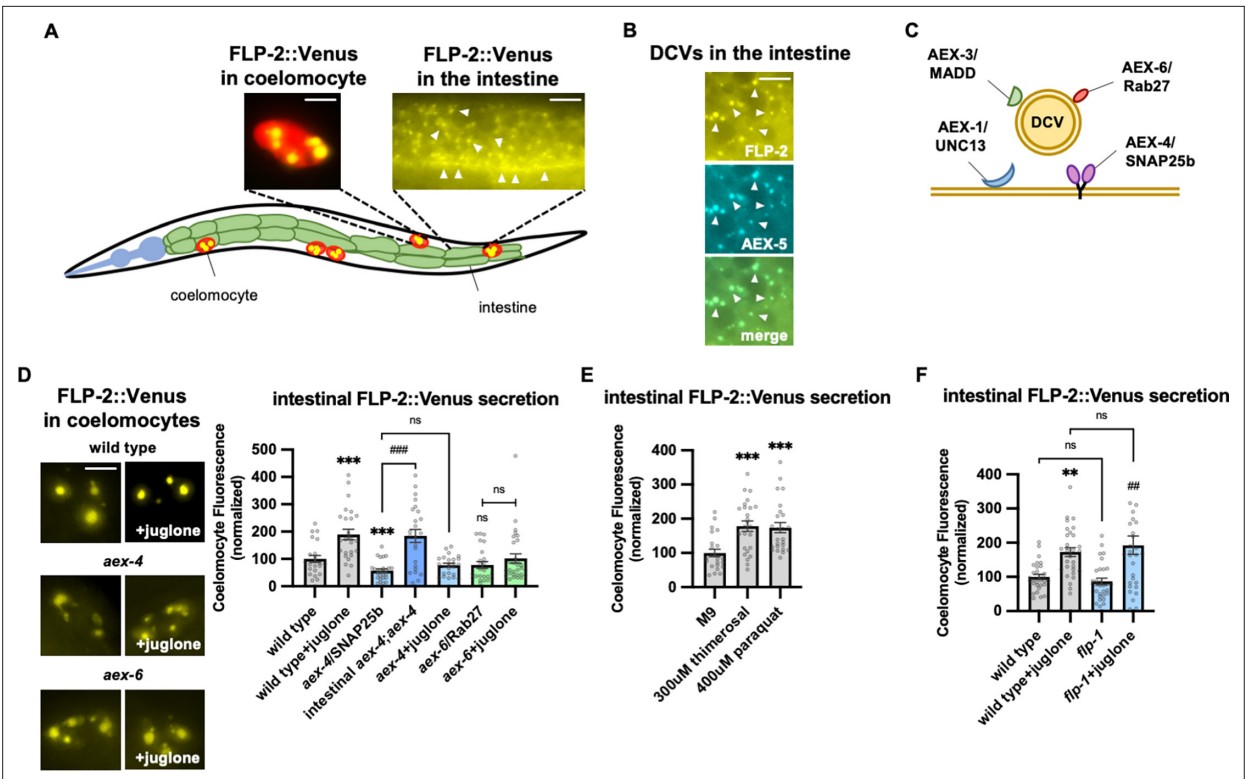

**Figure 2.** FLP-2 secretion from the intestine is stress regulated. (**A**) Schematic showing the positions of intestine and coelomocytes of transgenic animals co-expressing FLP-2::Venus in the intestine and mCherry in coelomocytes. Representative images of the posterior coelomocyte that have taken up Venus into the endocytic compartment (scale bar: 5 µM) and the posterior intestinal region showing the distribution of FLP-2::Venus in puncta in the intestine are shown (scale bar: 15 µM). (**B**) Representative images of fluorescence distribution in the posterior intestinal region of transgenic animals co-expressing FLP-2::Venus and AEX-5::mTur2 fusion proteins. Arrowheads denote puncta marked by both fusion proteins. Scale bar: 5 µM. (**C**) Schematic showing the locations of AEX-1/UNC13, AEX-3/MADD, AEX-4/SNAP25, and AEX-6/Rab27 relative to a dense core vesicle (DCV). (**D**) Representative images and quantification of average coelomocyte fluorescence of the indicated mutants expressing FLP-2::Venus fusion proteins in the intestine following M9 or 300 µM juglone for 10 min. Unlined *** and ns denote statistical significance compared to 'wild type'. n=29, 25, 24, 30, 23, 30, 25, 25, 25 independent animals. Scale bar: 5 µM. (**E**) Quantification of average coelomocyte fluorescence of transgenic animals expressing FLP-2::Venus fusion proteins in the intestine following treatment with M9 buffer or the indicated stressors for 10 min. Unlined *** denotes statistical significane compared to 'M9'. n=23, 25, 25 independent animals. (**F**) Quantification of average coelomocyte fluorescence of the indicated mutants expressing FLP-2::Venus fusion proteins in the intestine following M9 or 300 µM juglone treatment for 10 min. Unlined ** denotes statistical significance compared to 'wild type'; unlined ## denotes statistical significance compared to 'flp-1'; a denotes statistical significance compared to 'wild type+juglone'. n=30, 30, 30, 30 independent animals. (**D–F**) Data are mean values ± s.e.m. normalized to wild-type controls. ns, not significant, ** and ## $p<0.01$, *** and ### $p<0.001$ by Brown-Forsythe and Welch ANOVA with Dunnett's T3 multiple comparisons test.

The online version of this article includes the following source data and figure supplement(s) for figure 2:

**Source data 1.** Raw data used for plotting the figures.

**Figure supplement 1.** Specificity of juglone on intestinal peptide secretion, and FLP-2 and NLP-40 localization in the intestine.

**Figure supplement 1—source data 1.** Raw data used for plotting the figures.

## FLP-2 secretion from the intestine is $H_2O_2$-regulated

To directly investigate the mechanisms underlying the regulation of FLP-2 secretion, we examined FLP-2::Venus fusion proteins expressed in the intestine under various conditions (*Figure 2A*). FLP-2::Venus fusion proteins adopted a punctate pattern of fluorescence throughout the cytoplasm of intestinal cells and at the plasma membrane (*Figure 2A*), and FLP-2::Venus puncta co-localized with the DCV cargo protein AEX-5/PCSK5 tagged to mTurquoise2 (AEX-5::mTur2, *Figure 2B*). FLP-2::Venus fluorescence was also observed in the coelomocytes (marked by mCherry) (*Figure 2A and D*), indicating that FLP-2 is released from the intestine. SNAP25 forms a component of the core SNARE complex, which drives vesicular membrane fusion and transmitter release (*Chen and Scheller, 2001b*; *Goda, 1997*; *Jahn and Scheller, 2006*). *aex-4* encodes the *C. elegans* homolog of SNAP25, and mutations in *aex-4*

disrupt the secretion of neuropeptides from the intestine (*Lin-Moore et al., 2021*; *Mahoney et al., 2008*; *Wang et al., 2013*; *Figure 2C*). We found that *aex-4* null mutations significantly reduced coelomocyte fluorescence in FLP-2::Venus-expressing animals, and expression of *aex-4* cDNA selectively in the intestine fully restored FLP-2 secretion to *aex-4* mutants (*Figure 2D*). Together these results suggest that intestinal FLP-2 can be packaged into DCVs that undergo release via SNARE-dependent exocytosis.

To test whether intestinal FLP-2 secretion is regulated by oxidative stress, we examined coelomocyte fluorescence in FLP-2::Venus-expressing animals that had been exposed to a number of different commonly used oxidative stressors. We found that 10 min exposure to juglone, thimerosal, or paraquat, which promote mitochondria-targeted toxicity (*Castello et al., 2007*; *Elferink, 1999*; *Sharpe et al., 2012*), each significantly increased Venus fluorescence intensity in the coelomocytes compared to untreated controls (*Figure 2D and E*). We conducted four controls for specificity: First, juglone treatment did not significantly alter fluorescence intensity of mCherry expressed in coelomocytes (*Figure 2—figure supplement 1A*). Second, impairing intestinal DCV secretion by either *aex-4*/SNAP25 or *aex-6*/Rab27 mutations (*Lin-Moore et al., 2021*; *Mahoney et al., 2006*; *Mahoney et al., 2008*; *Thomas, 1990*) blocked the juglone-induced increase in coelomocyte fluorescence in FLP-2::Venus-expressing animals (*Figure 2C*). Third, *nlp-40* and *nlp-27* encode neuropeptide-like proteins that are released from the intestine but are not implicated in stress responses (*Liu et al., 2023*; *Taylor et al., 2021*; *Wang et al., 2013*). Juglone treatment had no detectable effects on coelomocyte fluorescence in animals expressing intestinal NLP-40::Venus or NLP-27::Venus fusion proteins (*Figure 2—figure supplement 1B and C*), and NLP-40::mTur2 puncta did not overlap with FLP-2::Venus puncta in the intestine (*Figure 2—figure supplement 1D*). Finally, *flp-1* mutants exhibited wild-type levels of FLP-2 secretion both in the absence and presence of juglone (*Figure 2E*). The distribution of FLP-2::Venus puncta in the intestine was not detectably altered by juglone treatment. Together, these results indicate that acute oxidative stress selectively increases the exocytosis of FLP-2-containing DCVs from the intestine, upstream of *flp-1* signaling.

## SOD-1 and SOD-3 superoxide dismutases regulate FLP-2 release

Juglone generates superoxide anion radicals (*Ahmad and Suzuki, 2019*; *Paulsen and Ljungman, 2005*) and juglone treatment of *C. elegans* increases ROS levels (*de Castro et al., 2004*) likely by promoting the global production of mitochondrial superoxide. Superoxide can then be rapidly converted into $H_2O_2$ by superoxide dismutase. To determine whether $H_2O_2$ impacts FLP-2 secretion, we first examined superoxide dismutase mutants. *C. elegans* encodes five superoxide dismutase genes (*sod-1* through *sod-5*). *sod-1* or *sod-3* null mutations blocked juglone-induced FLP-2 secretion without altering baseline FLP-2 secretion, whereas *sod-2*, *sod-4*, or *sod-5* mutations had no effect on FLP-2 secretion in the presence of juglone (*Figure 3A and B*, *Figure 3—figure supplement 1A*). *sod-1; sod-3* double mutants exhibited juglone-induced FLP-2 secretion defects that was similar to single mutants, without significantly altering FLP-2 secretion in the absence of stress (*Figure 3C*). *sod-1* encodes the ortholog of mammalian SOD1, which is a cytoplasmic SOD implicated in the development of amyotrophic lateral sclerosis and cancer (*Giglio et al., 1994*; *Papa et al., 2014*; *Wang et al., 2021*; *Zhang et al., 2007*). SOD-1::fusion proteins adopted a diffuse pattern of fluorescence in intestinal cells, consistent with a cytoplasmic localization (*Figure 3D*). Transgenes expressing the *sod-1* cDNA selectively in the intestine fully rescued the juglone-induced FLP-2::Venus secretion defects of *sod-1* mutants (*Figure 3A*). *sod-3* encodes a homolog of mammalian SOD2, which is a mitochondrial matrix SOD implicated in protection against oxidative stress-induced neuronal cell death (*Fukui and Zhu, 2010*; *Vincent et al., 2007*). Intestinal SOD-3::GFP fusion proteins localized to round structures that were surrounded by the outer membrane mitochondrial marker TOMM-20::mCherry (*Ahier et al., 2018*), consistent with a mitochondrial matrix localization (*Figure 3E*). Expression of *sod-3* cDNA in the intestine fully restored juglone-induced FLP-2 release to *sod-3* mutants (*Figure 3B*). *sod-3* variants lacking the mitochondrial localization sequence (*sod-3*(ΔMLS)) were no longer localized to mitochondria (*Figure 3F*) and failed to restore normal responsiveness to juglone to *sod-3* mutants (*Figure 3B*). Thus, the generation of $H_2O_2$ by either SOD-1 in the cytoplasm or by SOD-3 in the mitochondrial matrix is necessary for juglone to increase FLP-2 secretion.

Next, to determine if $H_2O_2$ can regulate FLP-2 secretion, we treated animals acutely with exogenous $H_2O_2$. We found that 10 min $H_2O_2$ treatment increased FLP-2::Venus secretion to a similar extent

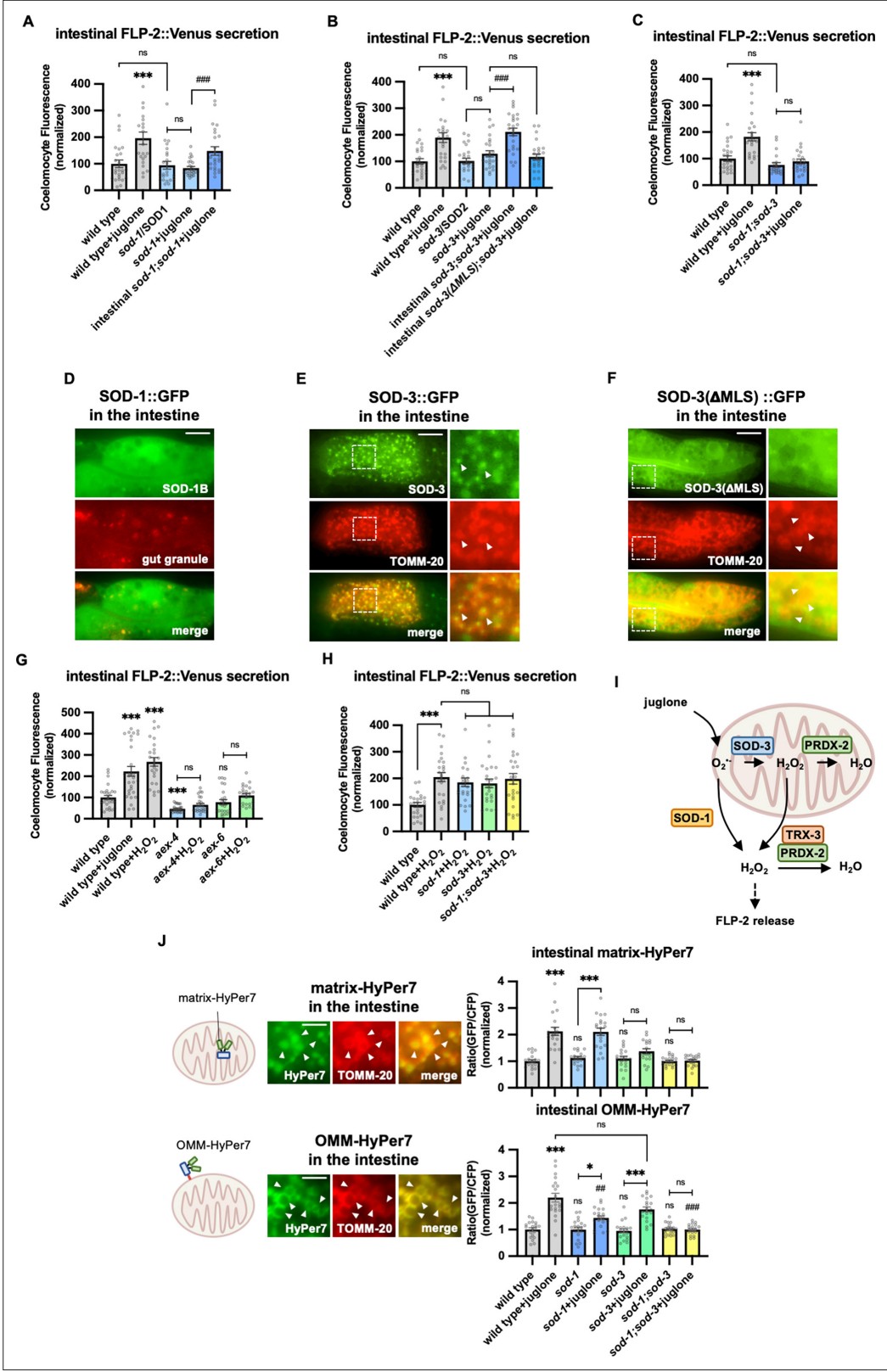

**Figure 3.** SOD-1/SOD-3 mediates endogenous $H_2O_2$ regulates FLP-2 release from the intestine. (**A**) Quantification of average coelomocyte fluorescence of the indicated mutants expressing FLP-2::Venus fusion proteins in the intestine following M9 or 300 μM juglone treatment for 10 min. Intestinal *sod*-1 denotes expression of *sod-1b* cDNA under the *ges-1* promoter. Unlined *** denotes statistical significance compared to 'wild type'. n=25,

*Figure 3 continued on next page*

*Figure 3 continued*

22, 24, 24, 25 independent animals. (**B**) Quantification of average coelomocyte fluorescence of the indicated mutants expressing FLP-2::Venus fusion proteins in the intestine following M9 or 300 µM juglone treatment for 10 min. Intestinal *sod-3* and *sod-3(ΔMLS)* denote intestinal expression of *sod-3* cDNA and *sod-3(ΔMLS)* variants, which lacks the mitochondrial localization sequence, under the *ges-1* promoter. Unlined *** denotes statistical significance compared to 'wild type'. n=25, 25, 25, 25, 25, 25 independent animals. (**C**) Quantification of average coelomocyte fluorescence of the indicated mutants expressing FLP-2::Venus fusion proteins in the intestine following M9 or 300 µM juglone treatment for 10 min. Unlined *** denotes statistical significance compared to 'wild type'. n=25, 25, 22, 25 independent animals. (**D**) Representative images of fluorescence distribution in the posterior intestinal region of transgenic animals expressing SOD-1b::GFP fusion proteins in contrast against autofluorescence of gut granules. Scale bar: 10 µM. (**E**) Representative images of fluorescence distribution in the posterior intestinal region of transgenic animals co-expressing SOD-3::GFP and TOMM-20::mCherry (to target mitochondria) fusion proteins. Scale bar: 15 µM. (**F**) Representative images of fluorescence distribution in the posterior intestinal region of transgenic animals co-expressing SOD-3(ΔMLS)::GFP and TOMM-20::mCherry fusion proteins. Scale bar: 15 µM. (**G**) Quantification of average coelomocyte fluorescence of the indicated mutants expressing FLP-2::Venus fusion proteins in the intestine following M9, 300 µM juglone, or 1 mM $H_2O_2$ treatment for 10 min. Unlined *** and ns denote statistical significance compared to 'wild type'. n=29, 30, 25, 25, 25, 24, 25 independent animals. (**H**) Quantification of average coelomocyte fluorescence of the indicated mutants expressing FLP-2::Venus fusion proteins in the intestine following M9 or 1 mM $H_2O_2$ treatment for 10 min. n=independent animals. (**I**) Schematic showing that SOD-1 and SOD-3 mediate juglone-induced $H_2O_2$ production in promoting FLP-2 release, and the PRDX-2/TRX-3 system detoxifies excessive $H_2O_2$. (**J**) Schematic, representative images and quantification of fluorescence in the posterior region of the indicated transgenic animals co-expressing mitochondrial matrix targeted HyPer7 (matrix-HyPer7) or mitochondrial outer membrane targeted HyPer7 (OMM-HyPer7) with TOMM-20::mCherry following M9 or 300 µM juglone treatment. Ratio of images taken with 500 nM (GFP) and 400 nM (CFP) for excitation and 520 nm for emission was used to measure $H_2O_2$ levels. Unlined *** and ns denote statistical significance compared to 'wild type'. Unlined ## and ### denote statistical significance compared to 'wild type+juglone'. (Top) n=20, 20, 18, 20, 19, 19, 20, 20 independent animals. (Bottom) n=20, 20, 19, 20, 20, 20, 20, 20 independent animals. Scale bar: 5 µM. (**A–C, G–H, and J**) Data are mean values ± s.e.m. normalized to wild-type controls. ns, not significant, * p<0.05, ## p<0.01, *** and ### p<0.001 by Brown-Forsythe and Welch ANOVA with Dunnett's T3 multiple comparisons test.

The online version of this article includes the following source data and figure supplement(s) for figure 3:

**Source data 1.** Raw data used for plotting the figures.

**Figure supplement 1.** SODs function in juglone-induced FLP-2 release from the intestine and mitochondrial mCherry control.

**Figure supplement 1—source data 1.** Raw data for plotting the figures.

---

as 10 min juglone treatment (Figue 3G). *aex-4*/SNAP25 or *aex-6*/Rab27 mutants exhibited no increase in FLP-2 secretion in response to $H_2O_2$ treatment compared to untreated controls (*Figure 3G*). In contrast, *sod-1* or *sod-3* mutants (or *sod-1; sod-3* double mutants) exhibited an increase in FLP-2 secretion in response to $H_2O_2$ that was similar to that of wild-type controls (*Figure 3H*), suggesting that exogenous $H_2O_2$ can bypass the requirement of SODs but not SNAREs to promote FLP-2 secretion. Together these results suggest that $H_2O_2$ generated by SODs can positively regulate intestinal FLP-2 exocytosis from DCVs (*Figure 3I*).

## SOD-1 and SOD-3 regulate intestinal mitochondrial $H_2O_2$ levels

To directly monitor $H_2O_2$ levels in the intestine, we generated transgenic animals in which HyPer7 was targeted to either the mitochondrial matrix (matrix-HyPer7) by generating fusion proteins with the cytochrome *c* MLS, or to the cytosolic face of the outer mitochondrial membrane (OMM-HyPer7) by generating fusion proteins with TOMM-20. When co-expressed in the intestine with the OMM marker TOMM-20::mCherry, matrix-HyPer7 formed round structures throughout the cytoplasm that were surrounded by the OMM, and OMM-HyPer7 formed ring-like structures throughout the cytoplasm that co-localized with the OMM marker (*Figure 3J*). Ten minute treatment with $H_2O_2$ significantly increased the fluorescence intensity by about twofold of both matrix-HyPer7 and OMM-HyPer7 without altering mitochondrial morphology or abundance (*Figure 3J*, *Figure 3—figure supplement 1B and C*), validating the utility of HyPer7 as a sensor for acute changes in $H_2O_2$ levels in and around intestinal mitochondria.

Juglone treatment for 10 min led to a similar twofold increase in matrix-HyPer7 fluorescence as $H_2O_2$ treatment (*Figure 3J*). *sod-3* mutations did not alter baseline $H_2O_2$ levels in the matrix, but they completely blocked juglone-induced increased $H_2O_2$ levels, whereas *sod-1* mutations had no effect on either baseline or juglone-induced increased $H_2O_2$ levels (*Figure 3J*). These results indicate that superoxide produced by juglone treatment is likely to be converted into $H_2O_2$ by SOD-3 in the mitochondrial matrix (*Figure 3I*).

Next, we examined $H_2O_2$ levels on the outer surface of mitochondria using OMM-HyPer7 and we found that juglone treatment led to a twofold increase in OMM-HyPer7 fluorescence, similar to $H_2O_2$ treatment (*Figure 3J*). *sod-3* or *sod-1* mutations did not alter baseline $H_2O_2$ levels on the OMM, but *sod-1* single mutations attenuated juglone-induced increases in OMM $H_2O_2$ levels, while *sod-3* mutations had no effect (*Figure 3J*). In *sod-1; sod-3* double mutants, the juglone-induced increase in OMM $H_2O_2$ levels was completely blocked, whereas baseline $H_2O_2$ levels in the absence of stress were unchanged (*Figure 3J*). These results suggest that *sod-3* and *sod-1* are exclusively required for $H_2O_2$ production by juglone and that both mitochondrial SOD-3 and cytosolic SOD-1 contribute to $H_2O_2$ levels in the cytosol. One model that could explain these results is that juglone-generated superoxide is converted into $H_2O_2$ both by SOD-3 in the matrix and by SOD-1 in the cytosol, and that the $H_2O_2$ generated in the matrix can exit the mitochondria to contribute to cytosolic $H_2O_2$ levels needed to drive FLP-2 secretion (*Figure 3I*).

## The peroxiredoxin-thioredoxin system regulates endogenous $H_2O_2$ levels and FLP-2 secretion

To determine whether endogenous $H_2O_2$ regulates FLP-2 secretion, we examined mutations in the peroxiredoxin-thioredoxin system. Peroxiredoxins and thioredoxins detoxify excessive $H_2O_2$ by converting it into water and they play a critical role in maintaining cellular redox homeostasis (*Netto and Antunes, 2016*; *Figure 4A*). *C. elegans* encodes two peroxiredoxin family members, *prdx-2* and *prdx-3*, that are expressed at high levels in the intestine (*Taylor et al., 2021*). Null mutations in *prdx-2* significantly increased FLP-2::Venus secretion compared to wild-type animals in the absence of stress (*Figure 4B*), whereas null mutations in *prdx-3* had no effect on FLP-2 secretion (*Figure 4—figure supplement 1A*). We observed a corresponding increase in both matrix-HyPer7 and OMM-HyPer7 fluorescence intensity in *prdx-2* mutants (*Figure 4C and D*), demonstrating that endogenous $H_2O_2$ is neutralized by peroxiredoxin and establishing a correlation between increased endogenous $H_2O_2$ levels and FLP-2 secretion. The increase in FLP-2 secretion in *prdx-2* mutants was not further increased by juglone treatment (*Figure 4B*). These results suggest that elevation in the levels of endogenously produced $H_2O_2$ in the intestine can positively regulate FLP-2 secretion.

There are three isoforms of *prdx-2* that arise by the use of alternative transcriptional start sites (*Figure 4A*). Expressing the *prdx-2b* isoform selectively in the intestine fully rescued the elevated FLP-2::Venus secretion defects of *prdx-2* mutants, whereas expressing *prdx-2a* or *prdx-2c* isoforms failed to rescue (*Figure 4B*, *Figure 4—figure supplement 1B and C*). To independently verify the role of *prdx-2b* function in FLP-2 release, we generated a *prdx-2b*-specific knockout mutant by introducing an in-frame stop codon within the *prdx-2b*-specific exon 1 using CRISPR/Cas9 (*prdx-2b(vj380)*; *Figure 4A*). *prdx-2b(vj380)* mutants exhibited increased $H_2O_2$ levels in the mitochondrial matrix and OMM (*Figure 4C and D*), as well as increased FLP-2::Venus secretion compared to wild-type controls that were indistinguishable from *prdx-2* null mutants (*Figure 4B*). *prdx-2b* mutations could no longer increase FLP-2 secretion when either *sod-1* or *sod-3* activity was impaired (*Figure 4E*, *Figure 4—figure supplement 1D*). Thus, the *prdx-2b* isoform normally inhibits FLP-2 secretion likely by promoting the consumption of $H_2O_2$ in the mitochondrial matrix and/or cytosol.

Once oxidized, peroxiredoxins are reduced by thioredoxins (TRXs) for reuse (*Netto and Antunes, 2016*). *trx-3* is an intestine-specific thioredoxin promoting protection against specific pathogen infections (*Jiménez-Hidalgo et al., 2014*; *Miranda-Vizuete et al., 2000*; *Netto and Antunes, 2016*). Mutations in *trx-3* elevated FLP-2::Venus release in the absence of juglone and expressing *trx-3* transgenes in the intestine restored wild-type FLP-2 release to *trx-3* mutants (*Figure 4B*). Juglone treatment failed to further enhance FLP-2::Venus release in *trx-3* mutants (*Figure 4B*). Mutations in cytoplasmic *sod-1* but not in mitochondrial *sod-3* reduced the elevated FLP-2::Venus release in *trx-3* mutants to wild-type levels (*Figure 4B*). Mutations in *trx-3* increased $H_2O_2$ levels in the OMM but had no effect on matrix $H_2O_2$ levels (*Figure 4C and D*). Thus, TRX-3 likely functions in the cytosol but not

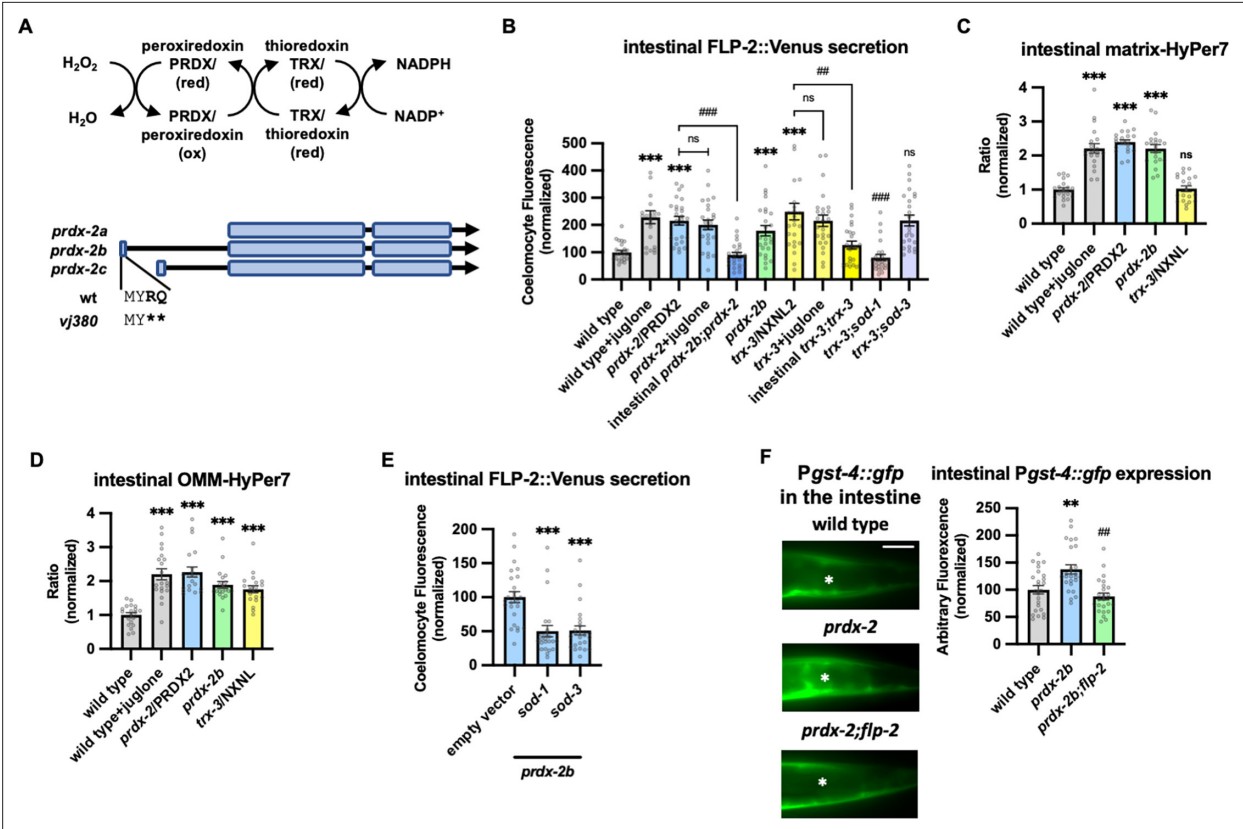

**Figure 4.** PRDX-2/PRDX and TRX-3/TRX regulate endogenous $H_2O_2$ and FLP-2 secretion. (**A**) (Top) Schematic showing the PRDX/TRX system in $H_2O_2$ detoxification. (Bottom) Schematic showing the three isoforms of *prdx-2* transcripts and *vj380* allele of *prdx-2b* knockout. (**B**) Quantification of average coelomocyte fluorescence of the indicated mutants expressing FLP-2::Venus fusion proteins in the intestine following M9 or 300 µM juglone treatment for 10 min. Intestinal *prdx-2b* denotes expression of *prdx-2b* cDNA under the *ges-1* promoter. Intestinal *trx-3* denotes expression of *trx-3* cDNA under the *ges-1* promoter. Unlined *** denotes statistical significance compared to 'wild type'; unlined ## and ### denote statistical significance compared to '*trx-3*'. n=25, 23, 25, 25, 25, 25, 25, 25, 25, 25, 25 independent animals. (**C and D**) Quantification of fluorescence in the posterior region of the indicated transgenic animals co-expressing matrix-HyPer7 (**C**) or OMM-HyPer7 (**D**) with TOMM-20::mCherry following M9 or 300 µM juglone treatment. Ratio of images taken with 500 nM (GFP) and 400 nM (CFP) for excitation and 520 nm for emission was used to measure $H_2O_2$ levels. Unlined *** and ns denote statistical significance compared to 'wild type'. (**C**) n=20, 20, 20, 20, 20 independent animals. (**D**) n=20, 20, 20, 20, 20 independent animals. (**E**) Quantification of average coelomocyte FLP-2::Venus fluorescence of transgenic animals fed with RNA interference (RNAi) bacteria targeting the indicated genes following M9 treatment for 10 min. Unlined *** denotes statistical significance compared to 'empty vector'. n=25, 23, 24 independent animals. (**F**) Representative images and quantification of average fluorescence in the posterior region of transgenic animals expressing P*gst-4::gfp* after 1 hr M9 or juglone exposure and 3 hr recovery. Asterisks mark the intestinal region for quantification. P*gst-4::gfp* expression in the body wall muscles, which appears as fluorescence on the edge animals in some images, was not quantified. Unlined ** denotes statistical significance compared to 'wild type', unlined ## denotes statistical analysis compared to '*prdx-2b*'. n=25, 25, 25 independent animals. Scale bar: 10 µM. (**B–F**) Data are mean values ± s.e.m. normalized to wild-type controls. ns, not significant, ** and ## p<0.01, *** and ### p<0.001 by Brown-Forsythe and Welch ANOVA with Dunnett's T3 multiple comparisons test.

The online version of this article includes the following source data and figure supplement(s) for figure 4:

**Source data 1.** Raw data for plotting the figures.

**Figure supplement 1.** PRDX-2 intestinal rescue and mediates SOD-3-dependent regulation of FLP-2 release.

**Figure supplement 1—source data 1.** Raw data for plotting the figures.

in the matrix to neutralize $H_2O_2$, and elevated $H_2O_2$ levels in the cytosol are sufficient to drive FLP-2 secretion without SOD-3-mediated $H_2O_2$ generation in the matrix.

Finally, to investigate the physiological significance of elevated endogenous $H_2O_2$ levels on the oxidative stress response, we examined the effects of *prdx-2b* mutations on expression of *gst-4*. *prdx-2b* mutants had significantly increased P*gst-4::gfp* expression in the intestine compared to wild-type controls (*Figure 4F*). The increased P*gst-4::gfp* expression in *prdx-2b* mutants was completely dependent upon *flp-2* signaling, since *gst-4* expression was reduced to wild-type levels in *prdx-2b*;

*flp-2* double mutants (*Figure 4F*). Together our data suggest that *prdx-2b* functions in the intestine to maintain redox homeostasis following SOD-1/SOD-3-mediated $H_2O_2$ production by regulating the secretion of FLP-2 (*Figure 3I*).

## PKC-2/PKCα/β mediates $H_2O_2$-induced FLP-2 secretion from the intestine

$H_2O_2$ functions as a cellular signaling molecule by oxidizing reactive cysteines to sulfenic acid, and this modification on target proteins can regulate intracellular signaling pathways (*García-Santamarina et al., 2014*). One of the validated targets of $H_2O_2$ signaling is the PKC family of serine threonine kinases (*Jia and Sieburth, 2021*; *Konishi et al., 1997*; *Konishi et al., 2001*; *Min et al., 1998*). *C. elegans* encodes four PKC family members including *pkc*-1 and *pkc*-2, which are expressed at highest levels in the intestine (*Islas-Trejo et al., 1997*; *Taylor et al., 2021*). *pkc-1* null mutants had no effect on baseline or juglone-induced FLP-2 secretion (*Figure 5—figure supplement 1A*). *pkc-2* null mutations did not alter baseline intestinal FLP-2 secretion, but they eliminated juglone-induced FLP-2 secretion (*Figure 5A*). *pkc-2* encodes a calcium and DAG stimulated PKCα/β PKC that regulates thermosensory behavior by promoting transmitter secretion (*Edwards et al., 2012*; *Land and Rubin, 2017*). Expressing *pkc-2* cDNA selectively in the intestine fully restored juglone-induced FLP-2 secretion to *pkc-2* mutants (*Figure 5A*), whereas expressing a catalytically inactive *pkc-2(K375R)* variant (*Van et al., 2021*) failed to rescue (*Figure 5A*). The intestinal site of action of *pkc-2* is in line with prior studies showing that *pkc-2* can function in the intestine to regulate thermosensory behavior (*Land and Rubin, 2017*). *pkc-2* mutants exhibited wild-type $H_2O_2$ levels in the mitochondrial matrix and OMM of the intestine in both the presence and absence of juglone (*Figure 5B and C*). Increasing $H_2O_2$ levels by either acute $H_2O_2$ treatment or by *prdx-2* mutation failed to increase FLP-2 secretion in *pkc-2* mutants (*Figure 5D and E*). To determine whether *pkc-2* can regulate the intestinal secretion of other peptides that are not associated with oxidative stress, we examined expulsion frequency, which is a measure of intestinal NLP-40 secretion (*Mahoney et al., 2008*; *Wang et al., 2013*). *pkc-2* mutants showed wild-type expulsion frequency (*Figure 5—figure supplement 1B*), indicating that intestinal NLP-40 release is largely unaffected. Together, these results show that *pkc-2* is not a general regulator of intestinal peptide secretion and instead functions downstream or in parallel to $H_2O_2$ to selectively promote FLP-2 secretion by a mechanism that involves phosphorylation of target proteins.

## DAG positively regulates FLP-2 secretion

PKCα/β family members contain two N-terminal C1 domains (C1A and C1B) whose binding to DAG promotes PKC recruitment to the plasma membrane (*Burns and Bell, 1991*; *Darby et al., 2017*; *Johnson et al., 2000*; *Kim et al., 2016*; *Ono et al., 1989*; *Yanase et al., 2011*). To address the role of DAG in promoting FLP-2 secretion by PKC-2, we examined mutants that are predicted to have altered DAG levels. Phosphatidylinositol phospholipase C beta (PLCβ) converts phosphatidyl inositol phosphate (PIP2) to DAG and inositol triphosphate (IP3, *Figure 6A*), and impairing PLC activity leads to reduced cellular DAG levels (*Nebigil, 1997*). *C. elegans* encodes two PLC family members whose expression is enriched in the intestine, *plc-2/* PLCβ and *egl-8/* PLCβ (*Taylor et al., 2021*). *plc-2* null mutants exhibited baseline and juglone-included FLP-2 secretion that were similar to wild-type controls (*Figure 6—figure supplement 1*). *egl-8* loss-of-function mutants exhibited wild-type baseline FLP-2 secretion, but juglone-induced FLP-2 secretion was completely blocked (*Figure 6B*). $H_2O_2$ levels in *egl-8* mutants were similar to wild-type controls, both in the presence and absence of juglone (*Figure 6C and D*). Thus, *egl-8/*PLCβ functions downstream of or in parallel to $H_2O_2$ production to promote FLP-2 secretion.

DAG kinase converts DAG into phosphatidic acid (PA), and is therefore a negative regulator or DAG levels (*Figure 6A*; *van Blitterswijk and Houssa, 2000*; *Topham, 2006*). In *C. elegans*, *dgk-2*/DGKε is the highest expressing DAG kinase in the intestine. Mutations in *dgk-2* elevated FLP-2::Venus secretion (*Figure 6E*) without altering $H_2O_2$ levels in the intestinal mitochondrial matrix or OMM (*Figure 6F and G*). Expressing *dgk-2* transgenes selectively in the intestine restored normal FLP-2::Venus secretion to *dgk-2* mutants (*Figure 6E*). Finally, the increase in FLP-2 secretion in *dgk-2* mutants was not further increased by juglone treatment, but it was completely blocked by *pkc-2* mutations or *aex-4/* SNAP25 mutations (*Figure 6E and H*). These results show that FLP-2 secretion can be regulated

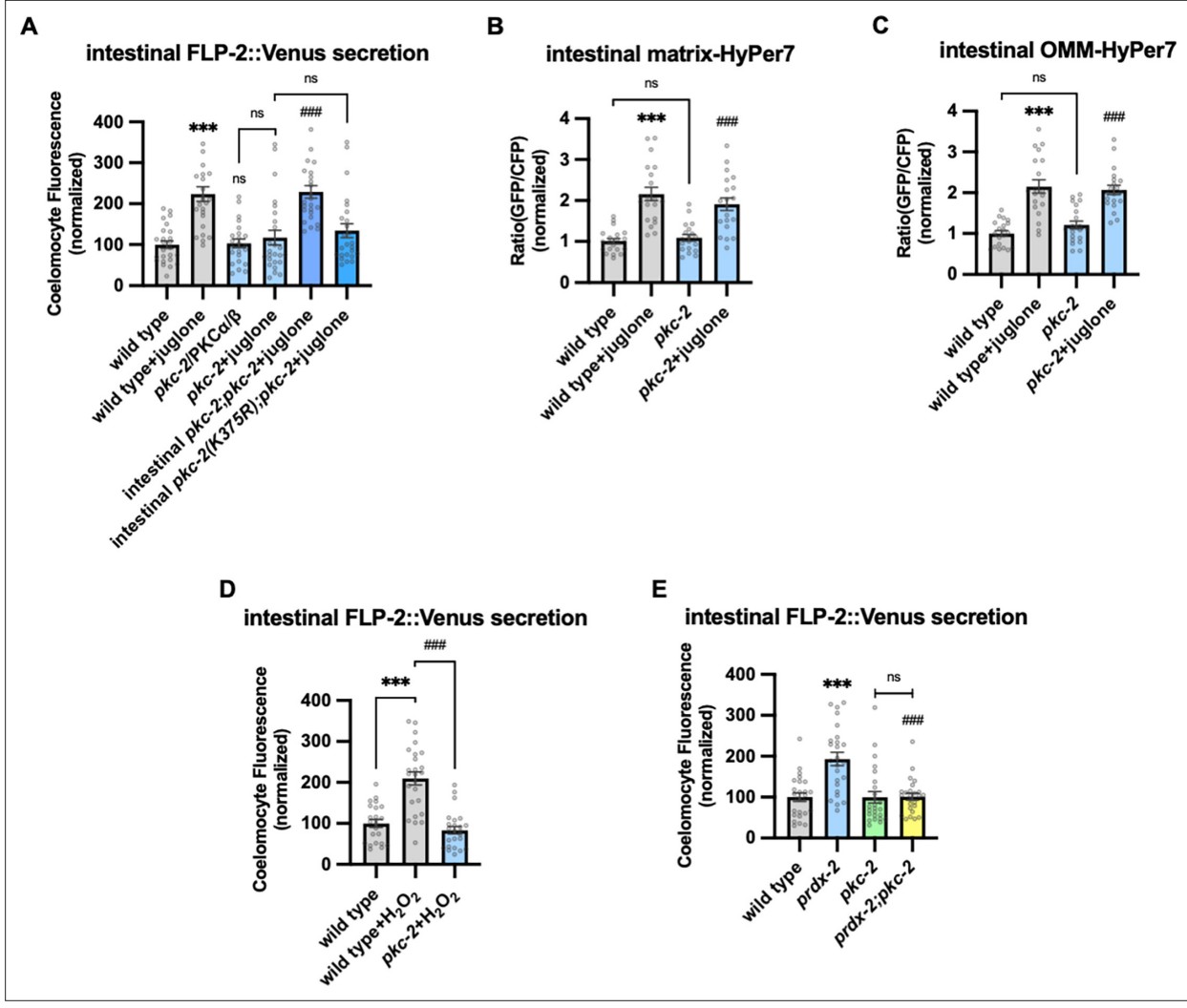

**Figure 5.** PKC-2/PKCα/β activation by H₂O₂ promotes FLP-2 secretion from the intestine. (**A**) Quantification of average coelomocyte fluorescence of the indicated mutants expressing FLP-2::Venus fusion proteins in the intestine following M9 or 300 µM juglone treatment for 10 min. Intestinal *pkc-2* denotes expression of *pkc-2b cDNA* under the ges-1 promoter. Intestinal *pkc-2b(K375R)* denotes expression of *pkc-2b(K375R)* variants under the *ges-1* promoter. Unlined *** and ns denote statistical significance compared to 'wild type'; ### denotes statistical significance compared to '*pkc*-2+juglone'. n=24, 24, 25, 25, 25, 25 independent animals. (**B and C**) Quantification of fluorescence in the posterior region of the indicated transgenic animals co-expressing matrix-HyPer7 (**B**) or OMM-HyPer7 (**C**) with TOMM-20::mCherry following M9 or 300 µM juglone treatment. Ratio of images taken with 500 nM (GFP) and 400 nM (CFP) for excitation and 520 nm for emission was used to measure H₂O₂ levels. Unlined *** denotes statistical significance compared to 'wild type'; unlined ### denotes statistical analysis compared to '*pkc-2*'. (**B**) n=20, 20, 19, 20 independent animals, (**C**) n=20, 20, 20, 20 independent animals. (**D**) Quantification of average coelomocyte fluorescence of the indicated mutants expressing FLP-2::Venus fusion proteins in the intestine following M9 or 1 mM H₂O₂ treatment for 10 min. n=23, 25, 25 independent animals. (**E**) Quantification of average coelomocyte fluorescence of the indicated mutants expressing FLP-2::Venus fusion proteins in the intestine following M9 treatment for 10 min. Unlined *** denotes statistical significance compared to 'wild type'; unlined ### denotes statistical significance compared to '*prdx-2*'. n=25, 25, 25, 25 independent animals. (**A–E**) Data are mean values ± s.e.m. normalized to wild-type controls. ns, not significant, *** and ### p<0.001 by Brown-Forsythe and Welch ANOVA with Dunnett's T3 multiple comparisons test.

The online version of this article includes the following source data and figure supplement(s) for figure 5:

**Source data 1.** Raw data for plotting the figures.

**Figure supplement 1.** Juglone promotes FLP-2 release in *pkc-1* mutants and expulsion analysis.

**Figure supplement 1—source data 1.** Raw data for plotting the figures.

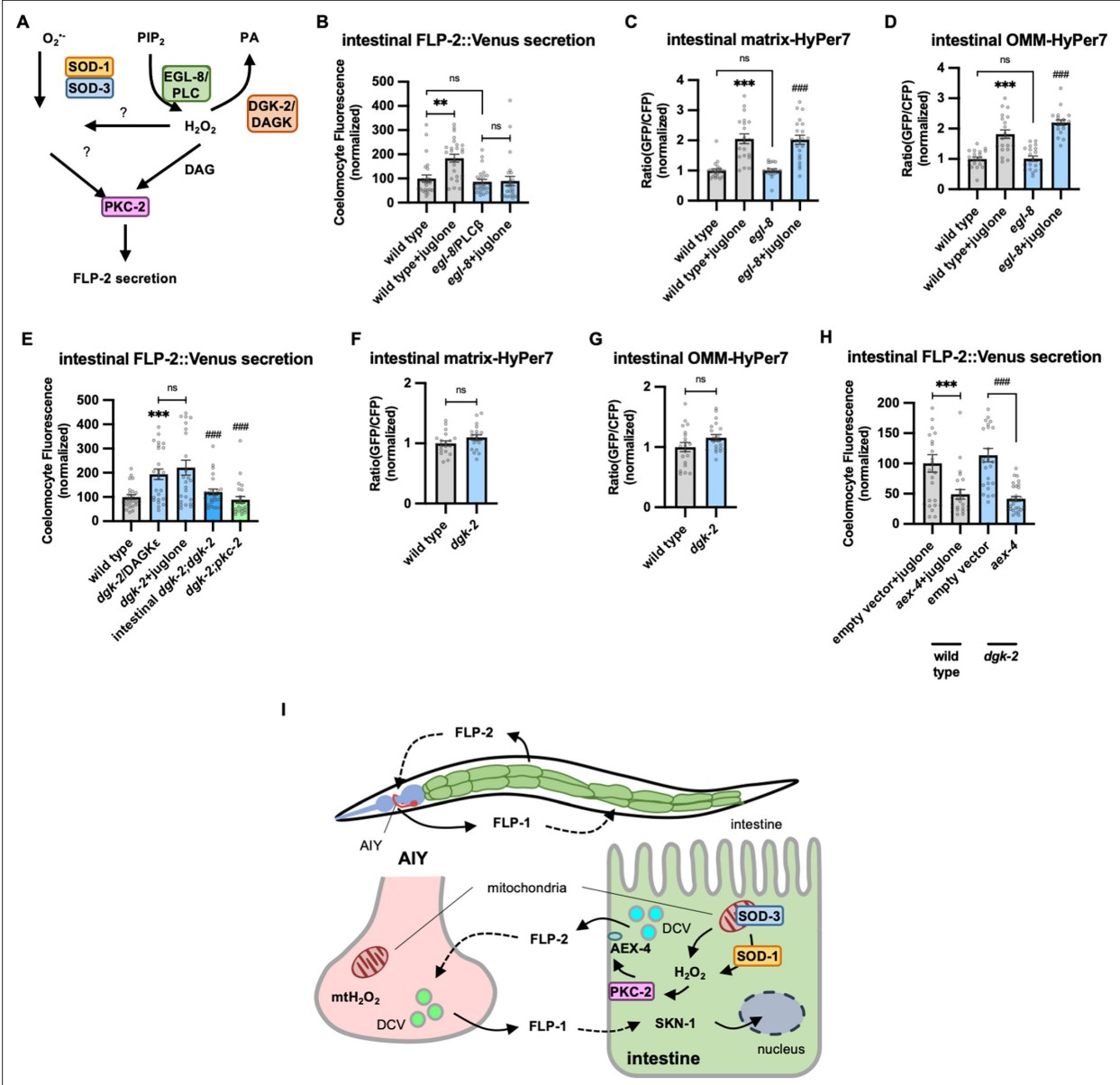

**Figure 6.** Diacylglycerol (DAG) promotes PKC-2-mediated FLP-2 secretion from the intestine. (**A**) Schematic showing PLC and DGK mediates DAG metabolism and DAG functions in $H_2O_2$-mediated FLP-2 signaling. (**B**) Quantification of average coelomocyte fluorescence of the indicated mutants expressing FLP-2::Venus fusion proteins in the intestine following M9 or juglone treatment for 10 min. n=25, 25, 25, 25 independent animals. (**C and D**) Quantification of fluorescence in the posterior region of the indicated transgenic animals co-expressing matrix-HyPer7 (**C**) or OMM-HyPer7 (**D**) with TOMM-20::mCherry following M9 or 300 µM juglone treatment. Ratio of images taken with 500 nM (GFP) and 400 nM (CFP) for excitation and 520 nm for emission was used to measure $H_2O_2$ levels. Unlined *** denotes statistical significance compared to 'wild type'; unlined ### denotes statistical significance compared to '*egl-8*'. (**C**) n=22, 20, 20, 21 independent animals, (**D**) n=20, 20, 20, 20 independent animals. (**E**) Quantification of average coelomocyte fluorescence of the indicated mutants expressing FLP-2::Venus fusion proteins in the intestine following M9 or 300 µM juglone treatment for 10 min. Intestinal *dgk-2* denotes expression of *dgk-2a* cDNA under the *ges-1* promoter. Unlined *** denotes statistical significance compared to 'wild type'; unlined ### denotes statistical significance compared to '*dgk-2*/DGKε'. n=25, 25, 25, 25, 24 independent animals. (**F and G**) Quantification of fluorescence in the posterior region of the indicated transgenic animals co-expressing matrix-HyPer7 (**F**) or OMM-HyPer7 (**G**) with TOMM-20::mCherry following M9 treatment. Ratio of images taken with 500 nM (GFP) and 400 nM (CFP) for excitation and 520 nm for emission was used to measure $H_2O_2$ levels. (**F**) n=20, 20 independent animals, (**G**) n=20, 20 independent animals. (**H**) Quantification of average coelomocyte fluorescence of the indicated transgenic animals fed with RNA interference (RNAi) bacteria targeting the indicated genes in the intestine following M9 treatment for 10 min. n=25, 24, 25, 30 independent animals. (**I**) (Top) Schematic showing the position of intestine and AIY neurons in FLP-1-FLP-2-mediated axis. (Bottom) Schematic showing endogenous $H_2O_2$ promotes PKC-2/AEX-4-mediated FLP-2 release from the intestine in FLP-1-FLP-2-regulated inter-tissue axis. (**B–H**) Data are

*Figure 6 continued on next page*

Figure 6 continued

mean values ± s.e.m. normalized to wild-type controls. (**B–E and H**) ns, not significant, ** p<0.01, *** and ### p<0.001 by Brown-Forsythe and Welch ANOVA with Dunnett's T3 multiple comparisons test. (**F and G**) ns, not significant by unpaired t test with Welch's correction.

The online version of this article includes the following source data and figure supplement(s) for figure 6:

**Source data 1.** Raw data for plotting the figures.

**Figure supplement 1.** Juglone promotes FLP-2 release in *plc-2* mutants.

**Figure supplement 1—source data 1.** Raw data for plotting the figures.

bidirectionally by DAG, and they suggest that DAG and $H_2O_2$ function in a common genetic pathway upstream of *pkc-2* to promote FLP-2 secretion (*Figure 6A*).

## Discussion

By screening for intercellular regulators of FLP-1 signaling from the nervous system in promoting the antioxidant response, we have uncovered a function for peptidergic signaling in mediating gut-to-neuron regulation of the antioxidant response in *C. elegans*. We identified the neuropeptide-like protein FLP-2 as an inter-tissue signal originating in the intestine to potentiate stress-induced FLP-1 release from AIY neurons and the subsequent activation of SKN-1 in the intestine. We found that $H_2O_2$ generated endogenously in the intestine or exogenously by acute oxidant exposure increases FLP-2 secretion from intestinal DCVs. $H_2O_2$ promotes FLP-2 exocytosis through PKC-2 and AEX-4/SNAP25. The use of oxidant-regulated peptide secretion exemplifies a mechanism that can allow the gut and the nervous system to efficiently and rapidly communicate through endocrine signaling to promote organism-wide protection in the face of intestinal stress (*Figure 6I*).

### A new function for *flp-2* signaling in the antioxidant response

Previous studies have identified roles for *flp-2* signaling in development and in stress responses. *flp-2* promotes locomotion during molting (*Chen et al., 2016*), promotes entry into reproductive growth (*Chai et al., 2022*), regulates longevity (*Kageyama et al., 2022*), and activates the UPR^mt cell non-autonomously during mitochondrial stress (*Shao et al., 2016*). The function we identified for *flp-2* in the antioxidant response has some notable similarities with *flp-2*'s other functions. First, *flp-2* mediates its effects at least in part by regulating signaling by other peptides. *flp-2* signaling increases the secretion of the neuropeptide like protein PDF-1 during lethargus (*Chen et al., 2016*) and INS-35/insulin-like peptide for its roles in reproductive growth choice and longevity (*Kageyama et al., 2022*), in addition to regulating AIY FLP-1 secretion (*Figure 5G*). Second, the secretion of FLP-2 is dynamic. FLP-2 secretion decreases during lethargus (*Chen et al., 2016*) and increases under conditions that do not favor reproductive growth (*Kageyama et al., 2022*), as well increasing in response to oxidants (*Figure 2C and D*). However, in some instances, the regulation of FLP-2 secretion may occur at the level of flp-2 expression (*Kageyama et al., 2022*), rather than at the level of exocytosis (*Figure 2C*). Finally, genetic analysis of *flp-2* has revealed that under normal conditions, *flp-2* signaling may be relatively low, since *flp-2* mutants show no defects in reproductive growth choice when animals are well fed (*Chai et al., 2022*), show only mild defects in locomotion during molting in non-sensitized genetic backgrounds (*Chen et al., 2016*), and do not have altered baseline FLP-1 secretion or antioxidant gene expression in the absence of exogenous oxidants (*Figure 1D and E*). It is notable that increased ROS levels are associated with molting (*Back et al., 2012*; *Knoefler et al., 2012*), aging (*Back et al., 2012*; *Van Raamsdonk and Hekimi, 2010*), starvation (*Tao et al., 2017*), and mitochondrial dysfunction (*Dingley et al., 2010*), raising the possibility that *flp-2* may be used in specific contexts associated with high ROS levels to affect global changes in physiology, behavior, and development.

One major difference we found for *flp-2* signaling in our study is that intestinal, but not neuronal *flp-2* activates the oxidative stress response, whereas *flp-2* originates from neurons for its reported roles in development and the UPR^mt. The intestine is uniquely poised to relay information about diet to the rest of the animal, and secretion of a number of neuropeptide-like proteins from the intestine (e.g. INS-11, PDF-2, and INS-7) is proposed to regulate responses to different bacterial food sources (*Lee and Mylonakis, 2017*; *Murphy et al., 2007*; *O'Donnell et al., 2018*). Since bacterial diet can impact ROS levels in the intestine (*Pang and Curran, 2014*), secretion of FLP-2 from the intestine

could function to relay information about bacterial diet to distal tissues to regulate redox homeostasis. In addition, the regulation of intestinal FLP-2 release by oxidants may meet a unique spatial, temporal, or concentration requirement for activating the antioxidant response that cannot be met by its release from the nervous system.

## AIY as a target for *flp-2* signaling

AIY interneurons receive sensory information from several neurons primarily as glutamatergic inputs to regulate behavior (*Chalasani et al., 2007*; *Clark et al., 2006*; *Satoh et al., 2014*). Our study reveals a previously undescribed mechanism by which AIY is activated through endocrine signaling originating from FLP-2 secretion from the intestine. FLP-2 could act directly on AIY, or it may function indirectly through upstream neurons that relay FLP-2 signals to AIY. *frpr-18* encodes an orexin-like GPCR that can be activated by FLP-2-derived peptides in transfected mammalian cells (*Larsen et al., 2013*; *Mertens et al., 2005*), and *frpr-18* functions downstream of *flp-2* in the locomotion arousal circuit (*Chen et al., 2016*). *frpr-18* is expressed broadly in the nervous system including in AIY (*Chen et al., 2016*), and loss-of-function *frpr-18* mutations lead to hypersensitivity to certain oxidants (*Ouaakki et al., 2023*). FRPR-18 is coupled to the heterotrimeric G protein Gαq (*Larsen et al., 2013*; *Mertens et al., 2005*), raising the possibility that FLP-2 may promote FLP-1 secretion from AIY by directly activating FRPR-18 in AIY. However, *flp-2* functions independently of *frpr-18* in the reproductive growth circuit, and instead functions in a genetic pathway with the GPCR *npr-30* (*Chai et al., 2022*). In addition, FLP-2-derived peptides (of which there are at least three) can bind to the GPCRs DMSR-1, or FRPR-8 in transfected cells (*Beets et al., 2023*). Identifying the relevant FLP-2 peptide(s), the FLP-2 receptor and its site of action will help to define the circuit used by intestinal *flp-2* to promote FLP-1 release from AIY.

FLP-1 release from AIY is positively regulated by $H_2O_2$ generated from mitochondria (*Jia and Sieburth, 2021*). Here, we showed that $H_2O_2$-induced FLP-1 release requires intestinal *flp-2* signaling. However, *flp-2* does not appear to promote FLP-1 secretion by increasing $H_2O_2$ levels in AIY (*Figure 1E*), and *flp-2* signaling is not sufficient to promote FLP-1 secretion in the absence of $H_2O_2$ (*Figure 1D*). These results point to a model whereby at least two conditions must be met in order for AIY to increase FLP-1 secretion: an increase in $H_2O_2$ levels in AIY itself, and an increase in *flp-2* signaling from the intestine. Thus AIY integrates stress signals from both the nervous system and the intestine to activate the intestinal antioxidant response through FLP-1 secretion. The requirement of signals from multiple tissues for FLP-1 secretion may function to limit the activation of SKN-1, since unregulated SKN-1 activation can be detrimental to organismal health (*Turner et al., 2024*). AIY shows a sporadic $Ca^{2+}$ response regardless of the presence of explicit stimulation (*Shimizu et al., 2019*; *Chalasani et al., 2007*; *Clark et al., 2006*), and FLP-1 secretion from AIY is calcium-dependent (*Jia and Sieburth, 2021*). How mitochondrial $H_2O_2$ levels are established in AIY by intrinsic or extracellular inputs, and how AIY integrates $H_2O_2$ and *flp-2* signaling to control FLP-1 secretion remain to be defined.

## A role for endogenous $H_2O_2$ in regulated neuropeptide secretion

Using HyPer7, we showed that acute juglone exposure results in a rapid elevation of endogenous $H_2O_2$ levels inside and outside intestinal mitochondria and a corresponding increase of FLP-2 release from the intestine that depends on the cytoplasmic superoxide dismutase *sod-1*, and mitochondrial *sod-3*. We favor a model whereby superoxide generated by juglone in the mitochondria is converted to $H_2O_2$ by SOD-3 in the matrix and by SOD-1 in the cytosol. In this case, both the superoxide generated by juglone and the $H_2O_2$ generated by SOD-3 would have to be able to exit the mitochondria and enter the cytosol. Superoxide and $H_2O_2$ can be transported across mitochondrial membranes through anion channels and aquaporin channels, respectively (*Bienert and Chaumont, 2014*; *Ferri et al., 2003*; *Han et al., 2003*; *Kontos et al., 1985*). The observation that both SOD-1 and SOD-3 activity are necessary to drive FLP-2 release suggests that $H_2O_2$ levels must reach a certain threshold in the cytoplasm to promote FLP-2 release, and this threshold requires the generation of $H_2O_2$ by both SOD-1 and SOD-3.

We identified a role for the antioxidant peroxiredoxin-thioredoxin system, encoded by *prdx-2* and *trx-3*, in maintaining low endogenous $H_2O_2$ levels in the intestine and in negatively regulating FLP-2 secretion. We showed that the *prdx-2b* isoform functions to inhibit FLP-2 secretion and to lower $H_2O_2$

levels in both the mitochondrial matrix and on the cytosolic side of mitochondria. These observations are consistent with a subcellular site of action for PRDX-2B in either the matrix only or in both the matrix and cytosol. In contrast, *trx-3* mutations do not alter mitochondrial $H_2O_2$ levels, suggesting that TRX-3 functions exclusively in the cytosol. Thus, the PRDX-2B-TRX-3 combination may function in the cytosol, and PRDX-2B may function with a different TRX family member in the matrix. There are several thioredoxin domain-containing proteins in addition to *trx-3* in the *C. elegans* genome (including *trx-5*/NXNL2) that could be candidates for this role. Alternatively, *prdx-2* may function alone or with other redox proteins. PRDX-2 may function without thioredoxins in its roles in light sensing and stress response in worms (*Li et al., 2016*; *Oláhová et al., 2008*; *Oláhová and Veal, 2015*). PRDX-2B contains a unique N-terminal domain that is distinct from the catalytic domain and is not found on the other PRDX-2 isoforms. This domain may be important for targeting PRDX-2B to specific subcellular location(s) where it can regulate FLP-2 secretion.

### Regulation of FLP-2 exocytosis by PKC-2/PKCα/β and AEX-4/SNAP25

We demonstrated that *pkc-2* mediates the effects of $H_2O_2$ on intestinal FLP-2 secretion, and $H_2O_2$- and DAG-mediated PKC-2 activation are likely to function in a common genetic pathway to promote FLP-2 secretion. Our observations that DAG is required for the effects of juglone (*Figure 6B*) are consistent with a two-step activation model for PKC-2, in which $H_2O_2$ could first modify PKC-2 in the cytosol, facilitating subsequent PKC-2 recruitment to the membrane by DAG. Alternatively, DAG could first recruit PKC-2 to membranes, where it is then modified by $H_2O_2$. We favor a model whereby $H_2O_2$ modification occurs in the cytosol, since $H_2O_2$ produced locally by mitochondria would have access to cytosolic pools of PKC-2 prior to its membrane translocation.

We defined a role for *aex-4*/SNAP25 in the fusion step of FLP-2 containing DCVs from the intestine under normal conditions as well as during oxidative stress. In neuroendocrine cells, phosphorylation of SNAP25 on Ser187 potentiates DCV recruitment into releasable pools (*Nagy et al., 2002*; *Shu et al., 2008*; *Yang et al., 2007*), and exocytosis stimulated by the DAG analog phorbol ester (*Gao et al., 2016*; *Shu et al., 2008*), without altering baseline SNAP25 function. Interestingly, the residue corresponding to Ser187 is conserved in AEX-4, raising the possibility that PKC-2 potentiates FLP-2 secretion by phosphorylating AEX-4. Since SNAP25 phosphorylation on Ser187 has been shown to increase its interaction with syntaxin and promote SNARE complex assembly in vitro (*Gao et al., 2016*; *Yang et al., 2007*), it is possible that elevated $H_2O_2$ levels could promote FLP-2 secretion by positively regulating SNARE-mediated DCV fusion at intestinal release sites on the basolateral membrane through AEX-4/SNAP25 phosphorylation by PKC-2. Prior studies have shown that PKC-2 phosphorylates the SNARE-associated protein UNC-18 in neurons to regulate thermosensory behavior (*Edwards et al., 2012*; *Land and Rubin, 2017*). Thus, PKC-2 may have multiple targets in vivo and target selection may be dictated by cell type and/or the redox status of the cell.

### Similar molecular mechanisms regulating FLP-1 and FLP-2 release

The molecular mechanisms we identified that regulate FLP-2 secretion from the intestine are similar in several respects to those regulating FLP-1 secretion from AIY. First, the secretion of both peptides is positively regulated by $H_2O_2$ originating from mitochondria. Second, in both cases, $H_2O_2$ promotes exocytosis of neuropeptide-containing DCVs by a mechanism that depends upon the kinase activity of PKC. Finally, the secretion of both peptides is controlled through the regulation of $H_2O_2$ levels by superoxide dismutases and by the peroxiredoxin-thioredoxin system. $H_2O_2$-regulated FLP-1 and FLP-2 secretion differ in the identity of the family members of some of the genes involved. *prdx-3-trx-2* and *sod-2* family members regulate $H_2O_2$ levels in AIY, whereas *prdx-2-trx-3* and *sod-1/sod-3* family members regulate $H_2O_2$ levels in the intestine. In addition, *pkc-1* promotes $H_2O_2$-induced FLP-1 secretion from AIY whereas *pkc-2* promotes $H_2O_2$-induced FLP-2 secretion from the intestine. Nonetheless, it is noteworthy that two different cell types utilize largely similar pathways for the $H_2O_2$-mediated regulation of neuropeptide release, raising the possibility that similar mechanisms may be utilized in other cell types and/or organisms to regulate DCV secretion in response to oxidative stress.

## Materials and methods

A complete list of *C. elegans* strains used in this study.

## Strains and transgenic lines

*C. elegans* strains were maintained at 20°C in the dark on standard nematode growth medium (NGM) plates seeded with OP50 *Escherichia coli* as food source, unless otherwise indicated. All strains were synchronized by picking mid L4 stage animal either immediately before treatment (for coelomocyte imaging and intestine imaging) or 24 hr before treatment (for P*gst-4::gfp* imaging). The wild-type strain was Bristol N2. Mutants used in this study were outcrossed at least four times.

Transgenic lines were generated by microinjecting plasmid mixes into the gonads of young adult animals following standard techniques (***Mello et al., 1991***). Microinjection mixes were prepared by mixing expression constructs with the co-injection markers pJQ70 (P*ofm-1::rfp*, 25 ng/µL), pMH163 (P*odr-1::mCherry*, 40 ng/µL), pMH164 (P*odr-1::gfp*, 40 ng/µL), or pDS806 (P*myo-3::mCherry*, 20 ng/µL) to a final concentration of 100 ng/µL. For tissue-specific expression, a 1.5 kb *rab-3* promoter was used for pan-neuronal expression (***Nonet et al., 1997***), a 2.0 kb *ges-1* or a 3.5 kb *nlp-40* promoter was used for intestinal expression (***Egan et al., 1995***; ***Wang et al., 2013***). At least three transgenic lines were examined for each transgene, and one representative line was used for quantification. Strains and transgenic lines used in this study are listed in ***Supplementary file 1***.

## Molecular biology

All gene expression vectors were constructed with the backbone of pPD49.26. Promoter fragments including P*rab-3* and P*ges-1* were amplified from genomic DNA; genes of interest, including cDNA fragments (*aex-5*, *snt-5*, *sod-1b*, *sod-3*, *isp-1*, *prdx-2a*, *prdx-2b*, *prdx-2c*, *trx-3*, *pkc-2b*, *dgk-2a*, *aex-4*) and genomic fragments (*flp-2*, *flp-40*, *nlp-36*, *nlp-27*), were amplified from cDNA library and genomic DNA respectively using standard molecular biology protocols. Expression plasmid of HyPer7 was designed based on reported mammalian expression plasmid for HyPer7 (***Pak et al., 2020***) and was synthesized by Thermo Fisher Scientific with codon optimization for gene expression in *C. elegans*. Plasmids and primers used in this study are listed in ***Supplementary file 1***.

## Toxicity assay

A stock solution of 50 mM juglone in DMSO was freshly made on the same day of liquid toxicity assay. 120 µM working solution of juglone in M9 buffer was prepared using stock solution before treatment. Between 60–80 synchronized adult animals were transferred into a 1.5 mL Eppendorf tube with fresh M9 buffer and washed three times, and a final wash was done with either the working solution of juglone with or M9 DMSO at the concentrations present in juglone-treated animals does not contribute to toxicity since DMSO treatment alone caused no significant change in survival compared to M9-treated controls (***Figure 1—figure supplement 1D***). Animals were incubated in the dark for 4 hr on rotating mixer before being transferred onto fresh NGM plates seeded with OP50 to recover in the dark at 20°C for 16 hr. Percentage of survival was assayed by counting the number of alive and dead animals. Toxicity assays were performed in triplicates.

## RNA interference

Plates for feeding RNAi were prepared as described (***Kamath and Ahringer, 2003***). Around 20–25 gravid adult animals with indicated genotype were transferred onto the RNAi plates that were seeded with HT115(DE3) bacteria transformed with L4440 vectors with targeted gene inserts or empty L4440 vectors. Eggs were collected for 4 hr to obtain synchronized populations. L4 stage animals were collected for further assays. RNAi clones were from Ahringer or Vidal RNAi library, or made from genomic DNA. Details were listed in ***Supplementary file 1***.

## Behavioral assays

The defecation motor program was assayed as previously described (***Liu and Thomas, 1994***). Twenty to thirty L4 animals were transferred onto a fresh NGM plate seeded with OP50 *E. coli* and were stored in a 20°C incubator for 24 hr. After 24 hr, 10 consecutive defecation cycles were observed from three independent animals and the mean and the standard error was calculated for each genotype. The pBoc and aBoc steps were recorded using custom Etho software (James Thomas Lab website: http://depts.washington.edu/jtlab/software/otherSoftware.html).

## Microscopy and fluorescence imaging

Approximately 30–40 age matched animals were paralyzed with 30 mg/mL 2,3-butanedione monoxime (BDM) in M9 buffer and mounted on 2% agarose pads. Images were captured using the Nikon eclipse 90i microscope equipped with Nikon Plan Apo ×20, ×40, ×60, and ×100 oil objective (NA=1.40), and a Photometrics Coolsnap ES2 camera or a Hamamatsu Orca Flash LT+CMOS camera. Metamorph 7.0 software (Universal Imaging/Molecular Devices) was used to capture serial image stacks and to obtain the maximum intensity projection image for analysis.

For transcriptional reporter imaging, young adult animals were transferred into a 1.5 mL Eppendorf tube with M9 buffer, washed three times and incubated in 50 µM working solution of juglone or M9 buffer control with equivalent DMSO for 1 hr in the dark on rotating mixer before recovering on fresh NGM plates with OP50 for 3 hr in the dark at 20°C. The posterior end of the intestine was imaged with the ×60 objective and quantification for average fluorescence intensity of a 16-pixel diameter circle in the posterior intestine was calculated using Metamorph.

For coelomocyte imaging, L4 stage animals were transferred in fresh M9 buffer on a cover slide, washed six times with M9 before being exposed to 300 µM juglone in M9 buffer (diluted from freshly made 50 mM stock solution), 1 mM $H_2O_2$ in M9 buffer, or M9 buffer. DMSO at the concentrations present in juglone-treated animals does not alter neuropeptide secretion since DMSO treatment alone caused no significant change in FLP-1::Venus or FLP-2::Venus coelomocyte fluorescence compared to M9-treated controls (*Figure 1—figure supplement 1D*, *Figure 2—figure supplement 1E*). Animals were then paralyzed in BDM and images of coelomocytes next to the posterior end of intestine were taken using the ×100 oil objective. Average fluorescence intensity of Venus from the endocytic compartments in the posterior coelomocytes was measured in ImageJ.

For fusion protein fluorescence imaging, L4 stage animals were exposed to M9 buffer or indicated oxidants for 10 min before being paralyzed in BDM and images taken of the posterior end of the intestine using ×100 oil objective. For HyPer7 imaging, Z stacks were obtained using GFP (excitation/emission: 500 nm/520 nm) and CFP (excitation/emission: 400 nm/520 nm) filter sets sequentially, HyPer7 fluorescence signal was quantified as the ratio of GFP to CFP fluorescence intensity changes with respect to the baseline [(Ft − F0)/F0].

## CRISPR/Cas9 editing

*prdx-2b(vj380)* knockout mutants were generated using a co-CRISPR protocol (*Arribere et al., 2014*). An sgRNA and a repair single-stranded oligodeoxynucleotides (ssODN) targeting *dpy-10* were co-injected with an sgRNA for genes of interest and an ssODN that induces homology-directed repair to introduce Cas9-mediated mutagenesis. Fifteen young adult animals were injected to produce around 30 singled F1 animals carrying Dpy or Rol phenotype. F2 animals were genotyped for mutations based on PCR and enzyme digest. Homozygous mutants were outcrossed with wild-type animals at least four times before being used for assays.

## Statistics

Statistical analysis was performed on GraphPad Prism 9. Unpaired t test with two tails was used for two groups and one-way ANOVA with multiple comparison corrections was used for three or more groups to determine the statistical significance. Statistical details and n are specified in the Figure legends. All comparisons are conducted based on wild-type controls unless indicated by lines between genotypes. Bar graphs with plots were generated using GraphPad Prism 9.

## Acknowledgements

*C. elegans* strains used in this study were provided by the Caenorhabditis Genetics Centre (CGC), which is funded by the NIH National Center for Research Resources (NCRR). We thank members of the Sieburth lab for critical reading and discussion of the manuscript. This work was supported by grants from National Institute of Health NINDS R01NS071085 and R01NS110730 to DS.

## Additional information

### Funding

| Funder | Grant reference number | Author |
|---|---|---|
| National Institute of Neurological Disorders and Stroke | R01NS071085 | Derek Sieburth |
| National Institute of Neurological Disorders and Stroke | R01NS110730 | Drew Young |

The funders had no role in study design, data collection and interpretation, or the decision to submit the work for publication.

### Author contributions

Qi Jia, Conceptualization, Data curation, Writing – original draft; Drew Young, Qixin Zhang, Data curation; Derek Sieburth, Conceptualization, Supervision, Funding acquisition, Writing – original draft, Writing – review and editing

### Author ORCIDs

Qi Jia https://orcid.org/0000-0002-8193-9786
Derek Sieburth https://orcid.org/0000-0002-1224-2758

Reviewer #1 (Public Review): https://doi.org/10.7554/eLife.97503.3.sa1
Reviewer #2 (Public Review): https://doi.org/10.7554/eLife.97503.3.sa2
Author response https://doi.org/10.7554/eLife.97503.3.sa3

## Additional files

### Supplementary files

- MDAR checklist
- Supplementary file 1. Strains, transgenic lines, and plasmids used in this study.

### Data availability

All data generated or analyzed during this study are included in the manuscript and source data files.

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

# Appendix 1

## Appendix 1—key resources table

| Reagent type (species) or resource | Designation | Source or reference | Identifiers | Additional information |
|---|---|---|---|---|
| Genetic reagent (*C. elegans*) | flp-1(ok2811) IV | CGC | OJ6555 | Mutant |
| Genetic reagent (*C. elegans*) | flp-2(ok3351) X | CGC | OJ5490 | Mutant |
| Genetic reagent (*C. elegans*) | flp-1(ok2811);flp-2(ok3351) | This paper | OJ10228 | Obtained from Derek Sieburth lab |
| Genetic reagent (*C. elegans*) | aex-4(sa22) X | CGC | OJ7466 | Mutant |
| Genetic reagent (*C. elegans*) | pkc-2(ok328) X | CGC | VC127 | Mutant |
| Genetic reagent (*C. elegans*) | vjIs150[pJQ60] | *Jia and Sieburth, 2021* | OJ3614 | FLP-1::Venus |
| Genetic reagent (*C. elegans*) | aex-5(sa23);vjIs150[pJQ60] | this paper | OJ5616 | Obtained from Derek Sieburth lab |
| Genetic reagent (*C. elegans*) | vjEx1748[pJQ298]; aex-5(sa23);vjIs150[pJQ60] | this paper | OJ5780 | Obtained from Derek Sieburth lab |
| Genetic reagent (*C. elegans*) | vjEx1753[pJQ299]; aex-5(sa23);vjIs150[pJQ60] | this paper | OJ5785 | Obtained from Derek Sieburth lab |
| Genetic reagent (*C. elegans*) | vjEx1753[pJQ299];aex-5(sa23); flp-2(ok3351);vjIs150[pJQ60] | this paper | OJ6334 | Obtained from Derek Sieburth lab |
| Genetic reagent (*C. elegans*) | flp-2(ok3351); vjIs150[pJQ60] | this paper | OJ5264 | Obtained from Derek Sieburth lab |
| Genetic reagent (*C. elegans*) | vjEx2882[pJQ366];f lp-2(ok3351);vjIs150[pJQ60] | this paper | OJ8818 | Obtained from Derek Sieburth lab |
| Genetic reagent (*C. elegans*) | vjEx2877[pJQ302]; flp-2(ok3511);vjIs150[pJQ60] | this paper | OJ8813 | Obtained from Derek Sieburth lab |
| Genetic reagent (*C. elegans*) | vjEx2877[pJQ302];vjIs150 | this paper | OJ10229 | Obtained from Derek Sieburth lab |
| Genetic reagent (*C. elegans*) | dvIs19[pAF15] | CGC | CL2166 | Obtained from Derek Sieburth lab |
| Genetic reagent (*C. elegans*) | flp-1(ok2811);dvIs19 | this paper | OJ2547 | Obtained from Derek Sieburth lab |
| Genetic reagent (*C. elegans*) | flp-2(ok3351);dvIs19 | this paper | OJ10230 | Obtained from Derek Sieburth lab |
| Genetic reagent (*C. elegans*) | flp-1(ok2811);flp-2(ok3511);dvIs19 | this paper | OJ6544 | Obtained from Derek Sieburth lab |
| Genetic reagent (*C. elegans*) | vjEx2877[pJQ302];dvIs19 | this paper | OJ10231 | Obtained in from Derek Sieburth lab |
| Genetic reagent (*C. elegans*) | vjEx2877[pJQ302]; flp-1(ok2281);dvIs19 | this paper | OJ10232 | Obtained from Derek Sieburth lab |
| Genetic reagent (*C. elegans*) | aex-1(sa9);vjIs150[pJQ60] | this paper | OJ5888 | Obtained from Derek Sieburth lab |
| Genetic reagent (*C. elegans*) | aex-3(js815);vjIs150[pJQ60] | this paper | OJ5890 | Obtained from Derek Sieburth lab |
| Genetic reagent (*C. elegans*) | aex-4(sa22);vIs150[pJQ60] | this paper | OJ5891 | Obtained from Derek Sieburth lab |
| Genetic reagent (*C. elegans*) | aex-6(sa24);vjIs150[pJQ60] | this paper | OJ5892 | Obtained from Derek Sieburth lab |

*Appendix 1 Continued on next page*

*Appendix 1 Continued*

| Reagent type (species) or resource | Designation | Source or reference | Identifiers | Additional information |
|---|---|---|---|---|
| Genetic reagent (*C. elegans*) | nlp-40(tm4085);vjIs150[pJQ60] | this paper | OJ5615 | Obtained from Derek Sieburth lab |
| Genetic reagent (*C. elegans*) | aex-2(sa3);vjIs150[pJQ60] | this paper | OJ5889 | Obtained from Derek Sieburth lab |
| Genetic reagent (*C. elegans*) | vjEx2035[pJQ305] | this paper | OJ6405 | Obtained from Derek Sieburth lab |
| Genetic reagent (*C. elegans*) | vjEx3069[pDY10]; vjEx2035[pJQ305] | this paper | OJ9469 | Obtained from Derek Sieburth lab |
| Genetic reagent (*C. elegans*) | aex-4(sa22);vjEx2035[pJQ305] | this paper | OJ6409 | Obtained from Derek Sieburth lab |
| Genetic reagent (*C. elegans*) | aex-6(sa24);vjEx2035[pJQ305] | this paper | OJ8345 | Obtained from Derek Sieburth lab |
| Genetic reagent (*C. elegans*) | flp-1(ok2811);vjEx2035[pJQ305] | this paper | OJ6641 | Obtained from Derek Sieburth lab |
| Genetic reagent (*C. elegans*) | vjIs40[pDS292] | this paper | OJ1002 | Obtained from Derek Sieburth lab |
| Genetic reagent (*C. elegans*) | vjEx3263[pJQ370] | this paper | OJ10237 | Obtained from Derek Sieburth lab |
| Genetic reagent (*C. elegans*) | vjEx3062[pDY14];vjEx2035[pJQ305] | this paper | OJ9567 | Obtained from Derek Sieburth lab |
| Genetic reagent (*C. elegans*) | sod-1(tm783);vjEx2035[pJQ305] | this paper | OJ9797 | Obtained from Derek Sieburth lab |
| Genetic reagent (*C. elegans*) | vjEx2814[pJQ419];sod-1 (tm783);vjEx2035[pJQ305] | this paper | OJ8588 | Obtained from Derek Sieburth lab |
| Genetic reagent (*C. elegans*) | sod-3(tm760);vjEx0235[pJQ305] | this paper | OJ8341 | Obtained from Derek Sieburth lab |
| Genetic reagent (*C. elegans*) | vjEx2910[pJQ389];sod-3 (tm760);vjEx2035[pJQ305] | this paper | OJ8933 | Obtained from Derek Sieburth lab |
| Genetic reagent (*C. elegans*) | vjEx2973[pJQ408];sod-3 (tm760);vjEx2035[pJQ305] | this paper | OJ9106 | Obtained from Derek Sieburth lab |
| Genetic reagent (*C. elegans*) | sod-1(tm783);sod-3(tm760); vjEx2035[pJQ305] | this paper | OJ10234 | Obtained from Derek Sieburth lab |
| Genetic reagent (*C. elegans*) | vjEx3266[pJQ420] | this paper | OJ10243 | Obtained from Derek Sieburth lab |
| Genetic reagent (*C. elegans*) | vjEx2993[pJQ407] | this paper | OJ9141 | Obtained from Derek Sieburth lab |
| Genetic reagent (*C. elegans*) | vjEx2996[pJQ409(Pges-1::sod-3(ΔMLS) cDNA::GFP)] | this paper | OJ9144 | Obtained from Derek Sieburth lab |
| Genetic reagent (*C. elegans*) | vjEx3020[pJQ383] | this paper | OJ9230 | Obtained from Derek Sieburth lab |
| Genetic reagent (*C. elegans*) | vjEx3014[pJQ411] | this paper | OJ9196 | Obtained from Derek Sieburth lab |
| Genetic reagent (*C. elegans*) | sod-1(tm783);vjEx3020[pJQ383] | this paper | OJ9281 | Obtained from Derek Sieburth lab |
| Genetic reagent (*C. elegans*) | sod-3(tm760);vjEx3020[pJQ383] | this paper | OJ9259 | Obtained from Derek Sieburth lab |
| Genetic reagent (*C. elegans*) | sod-1(tm783);sod-3(tm760); vjEx3020[pJQ383] | this paper | OJ10244 | Obtained from Derek Sieburth lab |
| Genetic reagent (*C. elegans*) | sod-1(tm783);vjEx3014[pJQ411] | this paper | OJ9795 | Obtained from Derek Sieburth lab |

*Appendix 1 Continued on next page*

*Appendix 1 Continued*

| Reagent type (species) or resource | Designation | Source or reference | Identifiers | Additional information |
|---|---|---|---|---|
| Genetic reagent (*C. elegans*) | sod-3(tm760);vjEx3014[pJQ411] | this paper | OJ9280 | Obtained from Derek Sieburth lab |
| Genetic reagent (*C. elegans*) | sod-1(tm783);sod-3(tm760); vjEx3014[pJQ411] | this paper | OJ10245 | Obtained from Derek Sieburth lab |
| Genetic reagent (*C. elegans*) | sod-2(ok1030);vjEx2035[pJQ305] | this paper | OJ10238 | Obtained from Derek Sieburth lab |
| Genetic reagent (*C. elegans*) | sod-4(gk101);vjEx2035[pJQ305] | this paper | OJ10239 | Obtained from Derek Sieburth lab |
| Genetic reagent (*C. elegans*) | sod-5(tm1146);vjEx2035[pJQ305] | this paper | OJ10240 | Obtained from Derek Sieburth lab |
| Genetic reagent (*C. elegans*) | prdx-2(gk169);vjEx2035[pJQ305] | this paper | OJ8991 | Obtained from Derek Sieburth lab |
| Genetic reagent (*C. elegans*) | prdx-2b(vj380);vjEx2035[pJQ305] | this paper | OJ10251 | Obtained from Derek Sieburth lab |
| Genetic reagent (*C. elegans*) | vjEx2926[pJQ381];prdx-2(gk169); vjEx2035[pJQ305] | this paper | OJ8996 | Obtained from Derek Sieburth lab |
| Genetic reagent (*C. elegans*) | trx-3(tm2820);vjEx2035[pJQ305] | this paper | OJ9249 | Obtained from Derek Sieburth lab |
| Genetic reagent (*C. elegans*) | vjEx3091[pJQ422];trx-3(tm2820); vjEx2035[pJQ305] | this paper | OJ9496 | Obtained from Derek Sieburth lab |
| Genetic reagent (*C. elegans*) | trx-3(tm2820);sod-1(tm783); vjEx2035[pJQ305] | this paper | OJ10252 | Obtained from Derek Sieburth lab |
| Genetic reagent (*C. elegans*) | trx-3(tm2820);sod-3(tm760); vjEx2035[pJQ305] | this paper | OJ10253 | Obtained from Derek Sieburth lab |
| Genetic reagent (*C. elegans*) | prdx-2(gk169); vjEx3020[pJQ383] | this paper | OJ9237 | Obtained from Derek Sieburth lab |
| Genetic reagent (*C. elegans*) | prdx-2b(vj380); vjEx3020[pJQ383] | this paper | OJ10247 | Obtained from Derek Sieburth lab |
| Genetic reagent (*C. elegans*) | trx-3(tm2820); vjEx3020[pJQ383] | this paper | OJ10249 | Obtained from Derek Sieburth lab |
| Genetic reagent (*C. elegans*) | prdx-2(gk169); vjEx3014[pJQ411] | this paper | OJ10246 | Obtained from Derek Sieburth lab |
| Genetic reagent (*C. elegans*) | prdx-2b(vj380); vjEx3014[pJQ411] | this paper | OJ10248 | Obtained from Derek Sieburth lab |
| Genetic reagent (*C. elegans*) | trx-3(tm2820); vjEx3014[pJQ411] | this paper | OJ10250 | Obtained from Derek Sieburth lab |
| Genetic reagent (*C. elegans*) | prdx-2b(vj380);dvIs19 | this paper | OJ10254 | Obtained from Derek Sieburth lab |
| Genetic reagent (*C. elegans*) | prdx-2b(vj380);flp-2(tm3351);dvIs19 | this paper | OJ10255 | Obtained from Derek Sieburth lab |
| Genetic reagent (*C. elegans*) | prdx-3(gk529);vjEx2035[pJQ305] | this paper | OJ10256 | Obtained from Derek Sieburth lab |
| *Genetic reagent (*C. elegans*) | vjEx3268[pJQ380];prdx-2(gk169); vjEx2035[pJQ305] | this paper | OJ10258 | Obtained from Derek Sieburth lab |
| Genetic reagent (*C. elegans*) | vjEx3270[pJQ399];prdx-2(gk169); vjEx2035[pJQ305] | this paper | OJ10260 | Obtained from Derek Sieburth lab |
| Genetic reagent (*C. elegans*) | prdx-2b(vj380);sod-3(tm760); vjEx2035[pJQ305] | this paper | OJ9250 | Obtained from Derek Sieburth lab |
| Genetic reagent (*C. elegans*) | pkc-2(ok328);vjEx2035[pJQ305] | this paper | OJ9682 | Obtained from Derek Sieburth lab |

*Appendix 1 Continued on next page*

*Appendix 1 Continued*

| Reagent type (species) or resource | Designation | Source or reference | Identifiers | Additional information |
|---|---|---|---|---|
| Genetic reagent (*C. elegans*) | vjEx2828[pJQ376];pkc-2(ok328); vjEx2035[pJQ305] | this paper | OJ8682 | Obtained from Derek Sieburth lab |
| Genetic reagent (*C. elegans*) | vjEx3131[pJQ446];pkc-2(ok328); vjEx2035[pJQ305] | this paper | OJ9657 | Obtained from Derek Sieburth lab |
| Genetic reagent (*C. elegans*) | pkc-2(ok328);vjEx3020[pJQ383] | this paper | OJ10279 | Obtained from Derek Sieburth lab |
| Genetic reagent (*C. elegans*) | pkc-2(ok328);vjEx3014[pJQ411] | this paper | OJ10280 | Obtained from Derek Sieburth lab |
| Genetic reagent (*C. elegans*) | prdx-2(gk169);pkc-2(ok328); vjEx2035[pJQ305] | this paper | OJ8939 | Obtained from Derek Sieburth lab |
| Genetic reagent (*C. elegans*) | pkc-1(nj3);vjEx2035[pJQ305] | this paper | OJ10278 | Obtained from Derek Sieburth lab |
| Genetic reagent (*C. elegans*) | egl-8(sa47);vjEx2035[pJQ305] | this paper | OJ9863 | Obtained from Derek Sieburth lab |
| Genetic reagent (*C. elegans*) | egl-8(sa47);vjEx3020[pJQ383] | this paper | OJ10281 | Obtained from Derek Sieburth lab |
| Genetic reagent (*C. elegans*) | egl-8(sa47);vjEx3014[pJQ411] | this paper | OJ10282 | Obtained from Derek Sieburth lab |
| *Genetic reagent (C. elegans)* | dgk-2(gk124);vjEx2035[pJQ305] | this paper | OJ10263 | Obtained from Derek Sieburth lab |
| Genetic reagent (*C. elegans*) | vjEx327[pJQ460];dgk-2(gk124); vjEx2035[pJQ305] | this paper | OJ10264 | Obtained from Derek Sieburth lab |
| Genetic reagent (*C. elegans*) | dgk-2(gk124);pkc-2(ok328); vjEx2035[pJQ305] | this paper | OJ10266 | Obtained from Derek Sieburth lab |
| Genetic reagent (*C. elegans*) | dgk-2(gk124);vjEx3020[pJQ383] | this paper | OJ10283 | Obtained from Derek Sieburth lab |
| Genetic reagent (*C. elegans*) | dgk-2(gk124);vjEx3014[pJQ411] | this paper | OJ10284 | Obtained from Derek Sieburth lab |
| Genetic reagent (*C. elegans*) | plc-2(ok1761);vjEx2035[pJQ305] | this paper | OJ9809 | Obtained from Derek Sieburth lab |
| Genetic reagent (*C. elegans*) | vjEx2936[pJQ382] | this paper | OJ9028 | Obtained from Derek Sieburth lab |
| Genetic reagent (*C. elegans*) | flp-2(ok3351);vjEx2936[pJQ382] | this paper | OJ10595 | Obtained from Derek Sieburth lab |
| Recombinant DNA reagent | Prab-3::aex-5 cDNA (plasmid) | this paper | pJQ298 | Obtained from Derek Sieburth lab |
| Recombinant DNA reagent | Pges-1::aex-5 cDNA | this paper | pJQ299 | Obtained from Derek Sieburth lab |
| Recombinant DNA reagent | Prab-3::flp-2 gDNA | this paper | pJQ366 | Obtained from Derek Sieburth lab |
| Recombinant DNA reagent | Pges-1::flp-2 gDNA | this paper | pJQ302 | Obtained from Derek Sieburth lab |
| Recombinant DNA reagent | Pges-1::aex-5::mTur2 | this paper | pDY10 | Obtained from Derek Sieburth lab |
| Recombinant DNA reagent | Pges-1::flp-2::Venus | this paper | pJQ305 | Obtained from Derek Sieburth lab |
| Recombinant DNA reagent | Pges-1::nlp-27 gDNA::Venus | this paper | pJQ370 | Obtained from Derek Sieburth lab |
| Recombinant DNA reagent | Pges-1::nlp-40::mTur2 | this paper | pDY14 | Obtained from Derek Sieburth lab |

*Appendix 1 Continued on next page*

*Appendix 1 Continued*

| Reagent type (species) or resource | Designation | Source or reference | Identifiers | Additional information |
|---|---|---|---|---|
| Recombinant DNA reagent | Pges-1::sod-1a cDNA | this paper | pJQ419 | Obtained from Derek Sieburth lab |
| Recombinant DNA reagent | Pges-1::sod-3 cDNA | this paper | pJQ389 | Obtained from Derek Sieburth lab |
| Recombinant DNA reagent | Pges-1::sod-3(ΔMLS) cDNA | this paper | pJQ408 | Obtained from Derek Sieburth lab |
| Recombinant DNA reagent | Pges-1::sod-1a cDNA::GFP | this paper | pJQ420 | Obtained from Derek Sieburth lab |
| Recombinant DNA reagent | Pges-1::sod-3 cDNA::GFP | this paper | pJQ407 | Obtained from Derek Sieburth lab |
| Recombinant DNA reagent | Pges-1::sod-3(ΔMLS) cDNA::GFP | this paper | pJQ409 | Obtained from Derek Sieburth lab |
| Recombinant DNA reagent | Pges-1::MLS::HyPer7 | this paper | pJQ383 | Obtained from Derek Sieburth lab |
| Recombinant DNA reagent | Pges-1::tomm-20::HyPer7 | this paper | pJQ411 | Obtained from Derek Sieburth lab |
| Recombinant DNA reagent | Pges-1::prdx-2b cDNA | this paper | pJQ381 | Obtained from Derek Sieburth lab |
| Recombinant DNA reagent | Pges-1::trx-3 cDNA | this paper | pJQ422 | Obtained from Derek Sieburth lab |
| Recombinant DNA reagent | Pges-1::prdx-2a cDNA | this paper | pJQ380 | Obtained from Derek Sieburth lab |
| Recombinant DNA reagent | Pges-1::prdx-2c cDNA | this paper | pJQ399 | Obtained from Derek Sieburth lab |
| Recombinant DNA reagent | Pges-1::pkc-2b cDNA | this paper | pJQ376 | Obtained from Derek Sieburth lab |
| Recombinant DNA reagent | Pges-1::pkc-2(K375R) cDNA | this paper | pJQ446 | Obtained from Derek Sieburth lab |
| Recombinant DNA reagent | Pges-1::dgk-2a cDNA | this paper | pJQ460 | Obtained from Derek Sieburth lab |
| Recombinant DNA reagent | Pttx-3::MLS::HyPer7 | this paper | pJQ382 | Obtained from Derek Sieburth lab |
| Sequence-based reagent | CCCCCCGCTAGCAAAAATGAAAT TAATTTTCCTGCTTTTGCTTTTTGG | this paper | aex-5_F | Obtained from Derek Sieburth lab |
| Sequence-based reagent | CCCCCCGGTACCTTATGA CATTGTTCCCACCACT | this paper | aex-5_R | Obtained from Derek Sieburth lab |
| Sequence-based reagent | CCCCGCTAGCAAAAATGCAA GTTTCTGGAATCCTATCTGC | this paper | flp-2_F | Obtained from Derek Sieburth lab |
| Sequence-based reagent | CCCCGGTACCTTATTGGA AGTCGTAATCTGGCAGC | this paper | flp-2_R | Obtained from Derek Sieburth lab |
| Sequence-based reagent | CCCCCCGGTACCTTATGA CATTGTTCCCACCACT | this paper | aex-5_R | Obtained from Derek Sieburth lab |
| Sequence-based reagent | CCCCACCGGTTTGGAAGT CGTAATCTGGCAGCGG | this paper | flp-2_R | Obtained from Derek Sieburth lab |
| Sequence-based reagent | CCCCGCTAGCAAAAATGATT TCCACTTCTTCACTTCTTATCCTT | this paper | nlp-27_F | Obtained from Derek Sieburth lab |
| Sequence-based reagent | CCCCACCGGTCTTTCC CCATCCACCGTATCC | this paper | nlp-27_R | Obtained from Derek Sieburth lab |
| Sequence-based reagent | CCCCGCTAGCAAAAATGTT TATGAATCTTCTCACTCAGGTCTCC | this paper | sod-1a_F | Obtained from Derek Sieburth lab |

*Appendix 1 Continued on next page*

*Appendix 1 Continued*

| Reagent type (species) or resource | Designation | Source or reference | Identifiers | Additional information |
|---|---|---|---|---|
| Sequence-based reagent | CCCCGGTACCTCACTGGGGAGCAGCGAGAG | this paper | sod-1a_R | Obtained from Derek Sieburth lab |
| Sequence-based reagent | CCCCGCTAGCAAAAATGCTGCAATCTACTGCTCGC | this paper | sod-3_F | Obtained from Derek Sieburth lab |
| Sequence-based reagent | CCCCGGTACCTTATTGTCGAGCATTGGCAAATCT | this paper | sod-3_R | Obtained from Derek Sieburth lab |
| Sequence-based reagent | CCCCGCTAGCAAAAATGAAGCACACTCTCCCAGA | this paper | sod-3(ΔMLS)_F | Obtained from Derek Sieburth lab |
| Sequence-based reagent | CCCCCCCGGGCTGGGGAGCAGCGAGAGCAA | this paper | sod-1_R | Obtained from Derek Sieburth lab |
| Sequence-based reagent | CCCCCCCGGGTTGTCGAGCATTGGCAAATCTC | this paper | sod-3_R | Obtained from Derek Sieburth lab |
| Sequence-based reagent | CCCCGCTAGCAAAAATGTATAGACAGATGTCGAAAGCATTC | this paper | prdx-2b_F | Obtained from Derek Sieburth lab |
| Sequence-based reagent | CCCCGGTACCTTAGTGCTTCTTGAAGTACTCTTGG | this paper | prdx-2a/b/c_R | Obtained from Derek Sieburth lab |
| Sequence-based reagent | CCCCGCTAGCAAAAATGGCTAAGAACTTTTTCTCCGGA | this paper | trx-3_F | Obtained from Derek Sieburth lab |
| Sequence-based reagent | CCCCGGTACCTTATGCACGGATTCTCTCGAGATT | this paper | trx-3_R | Obtained from Derek Sieburth lab |
| Sequence-based reagent | CCCCGCTAGCAAAAATGTCGAAAGCATTCATCGGAA | this paper | prdx-2a_F | Obtained from Derek Sieburth lab |
| Sequence-based reagent | CCCCGCTAGCAAAAATGTCTCTCGCTCCAAAGATG | this paper | prdx-2c_F | Obtained from Derek Sieburth lab |
| Sequence-based reagent | CCCCGCTAGCAAAAATGTCGTTGAGCACGAACAGC | this paper | pkc-2b_F | Obtained from Derek Sieburth lab |
| Sequence-based reagent | CCCCGATATCTCACGGTTCTACATCTTTGACATAAAAC | this paper | pkc-2b_R | Obtained from Derek Sieburth lab |
| Sequence-based reagent | ATTTCCTCACTGTTCTTGGAAGAGGATCGTTTG | this paper | pkc-2b(K375R)_F | Obtained from Derek Sieburth lab |
| Sequence-based reagent | ACACTTTTCCAAACGATCCTCTTCCAAGAACA | this paper | pkc-2b(K375R)_R | Obtained from Derek Sieburth lab |
| Sequence-based reagent | CCCCGCTAGCAAAAATGGAAATGGACGTGTATGATGAATTATTG | this paper | dgk-2a_F | Obtained from Derek Sieburth lab |
| Sequence-based reagent | CCCCGGTACCTTAGAAGAACATCCCACATCCGG | this paper | dgk-2a_R | Obtained from Derek Sieburth lab |
| Software, algorithm | Etho | James Thomas Lab | Defecation motor program analysis | http://depts.washington.edu/jtlab/software/otherSoftware.html |
| Software, algorithm | Metamorph 7.0 | Universal 709 Imaging/Molecular Devices | Image capture | |
| Software, algorithm | GraphPad Prism 9 | Prism | Statistical analysis | |

