## [Editor Report · eLife Assessment]

This study presents **convincing** evidence of the role of an intestine-released neuropeptide, FLP-2, in the oxidative stress response of *C. elegans*, as well as for the neural circuit pathway that regulates its release in response to sensing reactive oxygen species (i.e., H2O2). These **valuable** results advance the understanding of gut-brain signaling and the neural circuit basis of behavioral responses to stress.

---

## [Referee Report · Reviewer #1 (Public Review)]

Summary:

The main goal of the paper was to identify signals that activate FLP-1 release from AIY neurons in response to H2O2, previously shown by the authors to be an important oxidative stress response in the worm.

Strengths:

This study builds upon the authors' previous work (Jia and Sieburth 2021) by further elucidating the gut-derived signaling mechanisms that coordinate the organism-wide antioxidant stress response in *C. elegans*.

By detailing how environmental cues like oxidative stress are transduced into gut-derived peptidergic signals, this study represents a valuable advancement in understanding the integrated physiological responses governed by the gut-brain axis.

This work provides valuable mechanistic insights into the gut-specific regulation of the FLP-2 peptide signal.

Weaknesses:

Although the authors identify intestinal FLP-2 as the endocrine signal important for regulating the secretion of the neuronal antioxidant neuropeptide, FLP-1, there is no effort made to identify how FLP-2 levels regulate FLP-1 secretion or identify whether this regulation is occurring directly through the AIY neuron or indirectly. This is brought up in the discussion, but identifying a target for FLP-2 in this pathway seems like a crucial missing piece of information in characterizing this pathway.

Comments on revised version:

In general I think the revision is improved and addresses my comments. It is unfortunate though that the authors did not address my main question (did they test the frpr-18 mutant, and if not, why?). The fact that there are other potentially relevant receptors which bind to some FLP-2 peptides with low affinity is not really a justification not to test the known high-affinity receptor (i.e. FRPR-18).

---

## [Referee Report · Reviewer #2 (Public Review)]

Summary:

The core findings demonstrate that the neuropeptide-like protein FLP-2, released from the intestine of *C. elegans,* is essential for activating the intestinal oxidative stress response. This process is mediated by endogenous hydrogen peroxide (H2O2), which is produced in the mitochondrial matrix by superoxide dismutases SOD-1 and SOD-3. H2O2 facilitates FLP-2 secretion through the activation of protein kinase C family member pkc-2 and the SNAP25 family member aex-4. The study further elucidates that FLP-2 signaling potentiates the release of the antioxidant FLP-1 neuropeptide from neurons, highlighting a bidirectional signaling mechanism between the intestine and the nervous system.

Strengths:

This study presents a significant contribution to the understanding of the gut-brain axis and its role in oxidative stress response and significantly advances our understanding of the intricate mechanisms underlying the gut-brain axis's role in oxidative stress response. By elucidating the role of FLP-2 and its regulation by H2O2, the study provides insights into the molecular basis of inter-tissue communication and antioxidant defense in *C. elegans*. These findings could have broader implications for understanding similar pathways in more complex organisms, potentially offering new targets for therapeutic intervention in diseases related to oxidative stress and aging.

Weaknesses:

(1) The experimental techniques employed in the study were somewhat simple and could benefit from the incorporation of more advanced methodologies.

(2) The weak identification of the key receptors mediating the interaction between FLP-2 and AIY neurons, as well as the receptors in the gut that respond to FLP-1.

(3) The study could be improved by incorporating a sensor for the direct measurement of hydrogen peroxide levels.

Comments on revised version:

The authors answered my main questions. Although many of the experiments I suggested are in the beginning stages, it is clear that the authors noted that they are critical to understanding the mechanism of action of FLP-2, and hopefully they will continue to push forward and develop more approaches to further identify the receptor mechanism.

---

## [Author Response]

The following is the authors’ response to the original reviews.

**Public Reviews:**

**Reviewer #1 (Public Review):**
Summary:The main goal of the paper was to identify signals that activate FLP-1 release from AIY neurons in response to H2O2, previously shown by the authors to be an important oxidative stress response in the worm.Strengths:This study builds upon the authors' previous work (Jia and Sieburth 2021) by further elucidating the gut-derived signaling mechanisms that coordinate the organism-wide antioxidant stress response in *C. elegans.*By detailing how environmental cues like oxidative stress are transduced into gut-derived peptidergic signals, this study represents a valuable advancement in understanding the integrated physiological responses governed by the gut-brain axis.This work provides valuable mechanistic insights into the gut-specific regulation of the FLP2 peptide signal.Weaknesses:Although the authors identify intestinal FLP-2 as the endocrine signal important for regulating the secretion of the neuronal antioxidant neuropeptide, FLP-1, there is no effort made to identify how FLP-2 levels regulate FLP-1 secretion or identify whether this regulation is occurring directly through the AIY neuron or indirectly. This is brought up in the discussion, but identifying a target for FLP-2 in this pathway seems like a crucial missing piece of information in characterizing this pathway.

We agree that this is an important question. Specifically, identifying the FLP-2 receptor and its site of action is a major priority. Since there are at least four different receptors that have been functionally or physically linked to FLP-2 and there are at least three FLP-2 peptides, unraveling the components acting directly downstream of FLP-2 will require further investigation that we feel is beyond the scope of this current study. We have added a new panel (Fig 1E) addressing the requirements for flp-2 signaling on peroxide production in AIY. These results provide new mechanistic insight into how flp-2 impacts signaling in AIY and a new interpretation of these results has been added to the discussion.

**Reviewer #2 (Public Review):**
Summary:The core findings demonstrate that the neuropeptide-like protein FLP-2, released from the intestine of *C. elegans*, is essential for activating the intestinal oxidative stress response. This process is mediated by endogenous hydrogen peroxide (H2O2), which is produced in the mitochondrial matrix by superoxide dismutases SOD-1 and SOD-3. H2O2 facilitates FLP-2 secretion through the activation of protein kinase C family member pkc-2 and the SNAP25 family member aex-4. The study further elucidates that FLP-2 signaling potentiates the release of the antioxidant FLP-1 neuropeptide from neurons, highlighting a bidirectional signaling mechanism between the intestine and the nervous system.Strengths:This study presents a significant contribution to the understanding of the gut-brain axis and its role in oxidative stress response and significantly advances our understanding of the intricate mechanisms underlying the gut-brain axis's role in oxidative stress response. By elucidating the role of FLP-2 and its regulation by H2O2, the study provides insights into the molecular basis of inter-tissue communication and antioxidant defense in *C. elegans*. These findings could have broader implications for understanding similar pathways in more complex organisms, potentially offering new targets for therapeutic intervention in diseases related to oxidative stress and aging.Weaknesses:(1) The experimental techniques employed in the study were somewhat simple and could benefit from the incorporation of more advanced methodologies.

Thank you for your comment.

(2) The weak identification of the key receptors mediating the interaction between FLP-2 and AIY neurons, as well as the receptors in the gut that respond to FLP-1.

We agree that this is an important question. Specifically, identifying the FLP-2 receptor and its site of action is a major priority. Since there are at least four different receptors that have been functionally or physically linked to FLP-2 and there are at least three FLP-2 peptides, unraveling the components acting directly downstream of FLP-2 will require further investigation that we feel is beyond the scope of this current study.

(3) The study could be improved by incorporating a sensor for the direct measurement of hydrogen peroxide levels.

We have added a new panel (Fig 1E) addressing the requirements for flp-2 signaling on peroxide production in AIY using the genetically encoded peroxide sensor HyPer7. These results provide new mechanistic insight into how flp-2 impacts signaling in AIY and a new interpretation of these results has been added to the discussion. In addition, we have used HyPer7 to measure peroxide levels in the intestinal mitochondrial matrix and outer membrane (Figs 3, 4, 5, 6).

**Recommendations for the authors:**

**Reviewer #1 (Recommendations For The Authors):**
The major missing link in the study is how FLP-2 affects FLP-1 release from AIY: is the effect direct and does it require the previously described FLP-2 receptor FRPR-18? Although this possibility is discussed extensively (L511-528) so it is odd that the effect of an frpr-18 mutation was not tested (or if it was tested, why the results were not reported). If the authors haven't done this experiment (despite doing many less critical experiments) it would be good to know why.

We agree that this is an important question. Specifically, identifying the FLP-2 receptor and its site of action is a major priority. Since there are at least four different receptors that have been functionally or physically linked to FLP-2 and there are at least three FLP-2 peptides, unraveling the components acting directly downstream of FLP-2 will require further investigation that we feel is beyond the scope of this current study. We have added a new panel (Fig 1E) addressing the requirements for flp-2 signaling on peroxide production in AIY. These results provide new mechanistic insight into how flp-2 impacts signaling in AIY and a new interpretation of these results has been added to the discussion.

Results:

“To address how *flp-2* signaling regulates FLP-1 secretion from AIY, we examined H2O2 levels in AIY using a mitochondrially targeted pH-stable H2O2 sensor HyPer7 (mitoHyPer7, Pak et al. 2020). Mito-HyPer7 adopted a punctate pattern of fluorescence in AIY axons, and the average fluorescence intensity of axonal mito-HyPer7 puncta increased about two-fold following 10 minute juglone treatment (Fig 1E), in agreement with our previous studies using HyPer (Jia and Sieburth 2021), confirming that juglone rapidly increases mitochondrial AIY H2O2 levels. *flp-2* mutations had no significant effects on the localization or the average intensity of mito-HyPer7 puncta in AIY axons either in the absence of juglone, or in the presence of juglone (Fig 1E), suggesting that *flp-2* signaling promotes FLP-1 secretion by a mechanism that does not increase H2O2 levels in AIY. Consistent with this, intestinal overexpression of _flp-_2 had no effect on FLP-1::Venus secretion in the absence of juglone, but significantly enhanced the ability of juglone to increase FLP-1 secretion (Fig. 1D). We conclude that both elevated mitochondrial H2O2 levels and intact *flp-2* signaling from the intestine are necessary to increase FLP-1 secretion from AIY.”

More minor comments/suggestions:Line 172: No justification is given as to why the authors chose to focus on flp-2 over the other potential candidates identified in their RNAi screen.

We are currently examining the other neuropeptide hits from the screen, but we have no additional phenotypes to report.

Line 189: An explanation for the use of gDNA as opposed to cDNA should be given.

We have changed the text in the Results section as follows:

“Expressing a flp-2 genomic DNA (gDNA), fragment (containing both the flp-2a and flp-2b isoforms that arise by alternative splicing), specifically in the nervous system failed to rescue the FLP-1::Venus defects of flp-2 mutants, whereas expressing flp-2 selectively in the intestine fully restored juglone-induced FLP1::Venus secretion to flp-2 mutants (Fig. 1D).”

Line 249-253: nlp-40 and nlp-27 were not implicated in contributing to juglone toxicity in the RNAi screen performed previously by the authors, so it is unclear why both of these peptides are investigated beyond simply being released from the intestine. Confusingly, while Figure S2D shows no overlap between NLP-40 and FLP2, NLP-27 is omitted from the analysis.

We have clarified that these peptides are not implicated in stress responses, providing a clearer rational for why the serve as controls for specificity.

“Third, *nlp-40* and *nlp-27* encode neuropeptide-like proteins that are released from the intestine, but are not implicated in stress responses (Liu et al. 2023; Taylor et al. 2021; Wang et al. 2013), and juglone treatment had no detectable effects on coelomocyte fluorescence in animals expressing intestinal NLP-40::Venus or NLP-27::Venus fusion proteins (Fig. S2B and C), and NLP40::mTur2 puncta did not overlap with FLP-2::Venus puncta in the intestine (Fig. S2D).”

Line 262: A more detailed description of juglone's mechanism of action would be welcome here. Is juglone expected to act only in intestinal cells, or is its function more pervasive?

We have added more detail:

“Juglone generates superoxide anion radicals (Ahmad and Suzuki 2019; Paulsen and Ljungman 2005) and juglone treatment of *C. elegans* increases ROS levels (de Castro, Hegi de Castro, and Johnson 2004) likely by promoting the global production of mitochondrial superoxide. Superoxide can then be rapidly converted into H2O2 by superoxide dismutase.”

Line 414: Justification for why expulsion frequency is used here to quantify NLP-40 secretion is required, particularly because NLP-40::Venus was already used to quantify NLP-40 secretion via the coelomocyte fluorescence method in the experiments contributing to Figure S2.

We used expulsion frequency here because (1) it is an easier assay compared to the coelomocyte assay and (2) it is a functional assay. Defective NLP-40 exocytosis manifests as reduced exclusion frequency, therefore if NLP-40 secretion is defective in pkc-2 mutants, nlp-40 mutants should exhibit defects in expulsion frequency.

We have clarified this point:

“To determine whether *pkc-2* can regulate the intestinal secretion of other peptides that are not associated with oxidative stress, we examined expulsion frequency, which is a measure of NLP-40 secretion (Mahoney et al. 2008; Wang et al. 2013).”

Line 478: The discussion of neuronally-secreted kisspeptin in this context does not seem relevant as this paper has focused on intestinal peptide secretion.

We have removed this sentence:

In mammals, release of the RF-amide neuropeptide kisspeptin from the anteroventral periventricular nucleus (AVPV) regulates reproduction by inducing the release of gonadotropins via its stimulatory action on GnRH neurons (Han et al. 2005).

Line 526: DMSR-18 seems to be a typo. Possibly meant FRPR-8, as this is another FLP-2-activated GPCR identified in the screen (though notably, FRPR-8 is only activated by one of the two FLP-2 peptide products) On that note, DMSR-1 has two isoforms, and only one of them is activated by FLP-2 (and only one of the two FLP-2 peptides). This seems relevant to discuss.

We have corrected the text and we have added to the discussion the number of FLP-2 peptides:

“In addition, certain FLP-2-derived peptides (of which there are at least three) can bind to the GPCRs DMSR-1, or FRPR-8 in transfected cells (Beets et al. 2023). Identifying the relevant FLP-2 peptide(s), the FLP-2 receptor and its site of action will help to define the circuit used by intestinal *flp-2* to promote FLP-1 release from AIY.”

Line 534: An explanation or speculation into why this integration might be necessary would be welcome here.

We have edited this paragraph:

“FLP-1 release from AIY is positively regulated by H2O2 generated from mitochondria (Jia and Sieburth 2021). Here we showed that H2O2-induced FLP-1 release requires intestinal *flp-2* signaling. However, *flp-2* does not appear to promote FLP-1 secretion by increasing H2O2 levels in AIY (Fig 1E), and *flp-2* signaling is not sufficient to promote FLP-1 secretion in the absence of H2O2 (Fig. 1D). These results point to a model whereby at least two conditions must be met in order for AIY to increase FLP-1 secretion: an increase in H2O2 levels in AIY itself, and an increase in *flp-2* signaling from the intestine. Thus AIY integrates stress signals from both the nervous system and the intestine to activate the intestinal antioxidant response through FLP-1 secretion. The requirement of signals from multiple tissues for FLP-1 secretion may function to limit the activation of SKN-1, since unregulated SKN-1 activation can be detrimental to organismal health (Turner, Ramos, and Curran 2024).”

Line 569: Should specify what these candidates are.

There are 11 proteins with thioredoxin fold domains. We modified the sentence to list one of them.

“There are several thioredoxin-domain containing proteins in addition to *trx-3* in the *C. elegans* genome that could be candidates for this role (e.g. *trx-5* and others).”

Line 660: Details about whether the M9 control had an equivalent amount of DMSO as the juglone+M9 condition is required.

We have performed toxicity assay and neuropeptide release assays comparing M9 DMSO, and Juglone treatment and we have included this new data in Fig S1C, D and S2E. Methods:

“A stock solution of 50mM juglone in DMSO was freshly made on the same day of liquid toxicity assay. 120μM working solution of juglone in M9 buffer was prepared using stock solution before treatment. Around 60-80 synchronized adult animals were transferred into a 1.5mL Eppendorf tube with fresh M9 buffer and washed three times, and a final wash was done with either the working solution of juglone with or M9 DMSO at the concentrations present in juglone-treated animals does not contribute to toxicity since DMSO treatment alone caused no significant change in survival compared to M9-treated controls (Fig. S1C).

For coelomocyte imaging, L4 stage animals were transferred in fresh M9 buffer on a cover slide, washed six times with M9 before being exposed to 300μM juglone in M9 buffer (diluted from freshly made 50mM stock solution), 1mM H2O2 in M9 buffer, or M9 buffer. DMSO at the concentrations present in juglone-treated animals does not alter neuropeptide secretion since DMSO treatment alone caused no significant change in FLP-1::Venus or FLP-2::Venus coelomocyte fluorescence compared to M9-treated controls. (Fig. S1D and S2E).”

Line 1191: Should be FLP-1:Venus in AIY, not the intestine

Corrected.

In general, the significance of reporting in the figures is very unclear. "a, b, c" to report statistical analysis is confusing in the figure legends, and also unnecessary when they denote non-significance. There are some cases where it is reported that a symbol (eg. ***) denotes statistical significance, but there is no indication of what level of statistical significance the symbol represents (for example, in Figures 2C and 2D)

Levels of significance was summarized in the end of legend for each figure unless indicated for specific symbols (for example Fig. 1C), we have edited this figure legend:

“E Representative images and quantification of fluorescence of matrix-targeted HyPer7 in the axon of AIY following M9 or juglone treatment for 10min. Arrowheads denote puncta marked by MLS::HyPer7 fusion proteins (Excitation: 500 and 400nm; emission: 520nm). Ratio of images taken with 500nM (GFP) and 400nM (CFP) for excitation was used to measure H2O2 levels. Unlined *** and ns denote statistical analysis compared to “wild type”. n = 25, 25, 25, 25 independent animals. Scale bar: 10μM.

F Representative images and quantification of average fluorescence in the posterior region of transgenic animals expressing P_gst-4::gfp_ after 4h vehicle M9 or juglone exposure. Asterisks mark the intestinal region used for quantification. P_gst-4::gfp_ expression in the body wall muscles, which appears as fluorescence on the edge animals in some images, was not quantified. Unlined *** and ns denote statistical analysis compared to “wild type”; unlined ## and ### denotes statistical analysis compared to “wild type+juglone”. n = 25, 26, 25, 25, 25, 25, 25, 25 independent animals. Scale bar: 10μM.”

Figure 2C: It is unclear which conditions have H2O2 treatment (as described in the legend). There is also no mention of what ### indicates.

Levels of significance for ### was summarized in the end of legend, No H2O2 treatment was performed in this assay, we have edited this figure legend:

“C. Representative images and quantification of average coelomocyte fluorescence of the indicated mutants expressing FLP-2::Venus fusion proteins in the intestine following M9 or juglone treatment for 10min. Unlined *** and ns denote statistical analysis compared to “wild type”. n = 29, 25, 24, 30, 23, 30, 25, 25, 25 independent animals. Scale bar: 5μM.”

Figure 2D: It is not previously mentioned that M9 condition contains DMSO, as implied by the legend.

We have edited this figure legend:

“D. Quantification of average coelomocyte fluorescence of transgenic animals expressing FLP-2::Venus fusion proteins in the intestine following treatment of fresh M9 buffer or the indicated stressors for 10min. Unlined *** denotes statistical analysis compared to “M9”. n = 23, 25, 25 independent animals.”

Figure 3J: The y-axis label should more clearly describe the ratio being measured.

We have updated the panel and this figure legend:

“J. Schematic, representative images and quantification of fluorescence in the posterior region of the indicated transgenic animals co-expressing mitochondrial matrix targeted HyPer7 (matrix-HyPer7) or mitochondrial outer membrane targeted HyPer7 (OMMHyPer7) with TOMM-20::mCherry following M9 juglone or H2O2 treatment. Ratio of images taken with 500nM (GFP) and 400nM (CFP) for excitation and 520nm for emission was used to measure H2O2 levels. Unlined *** and ns denote statistical analysis compared to “wild type; unlined ## denotes statistical analysis compared to “wild type+juglone”. (top) n = 20, 20, 18, 20, 19, 19, 20, 20 independent animals. (Bottom) n = 20, 20, 19, 20, 20, 20, 20, 20 independent animals. Scale bar: 5μM.”

Figure S3A: *** is mislabelled. It should be a comparison to wildtype.

We have edited this figure legend:

“A. Quantification of average coelomocyte fluorescence of the indicated mutants expressing FLP-2::Venus fusion proteins in the intestine following M9 or juglone treatment for 10min. Unlined *** denotes statistical analysis compared to “wild type”; ### and ns denote statistical analysis compared to “wild type+juglone”. n = 29, 27, 29, 27, 25, 26, 24 independent animals.”

**Reviewer #2 (Recommendations For The Authors):**
(1) The localization experiments could benefit from the application of ultra-high-resolution fluorescence microscopy. This would allow for a more detailed analysis of the spatial distribution of SOD-1/3::GFP in relation to mitochondria-targeted TOMM-20::mCherry fusion proteins in the posterior intestinal region of transgenic animals.

We agree that high resolution microscopy would be a great way to more precisely localize SOD proteins relative to the mitochondria, and this would enhance understanding of the source of peroxide in this system. We do not conduct this type of microcopy in the lab, so this approach would require a collaboration with a lab that is set up for this. Thus we feel that this is beyond the scope of the current study.

(2) The paper may note the challenge of directly measuring mitochondrial H2O2 concentrations. However, advancements in chemical or fluorescent sensors for H2O2 detection within mitochondria could provide more direct evidence of its role in FLP-2 secretion.

We have considered using chemical sensors, but many are either not efficiently taken up by worms (the skin is largely impermeable to all but the most hydrophobic molecules), or they would label peroxide indiscriminately in all tissues making detection specifically in the intestine challenging. We have had good luck with genetically encoded peroxide sensors since they provide tissue specificity and good spatial resolution depending on where we target them. We have added imaging results for HyPer7 in the AIY neuron to Figure 1E.

Results:

“To address how *flp-2* signaling regulates FLP-1 secretion from AIY, we examined H2O2 levels in AIY using a mitochondrially targeted pH-stable H2O2 sensor HyPer7 (mitoHyPer7, Pak et al. 2020). Mito-HyPer7 adopted a punctate pattern of fluorescence in AIY axons, and the average fluorescence intensity of axonal mito-HyPer7 puncta increased about two-fold following 10 minute juglone treatment (Fig 1E), in agreement with our previous studies using HyPer (Jia and Sieburth 2021), confirming that juglone rapidly increases mitochondrial AIY H2O2 levels. *flp-2* mutations had no significant effects on the localization or the average intensity of mito-HyPer7 puncta in AIY axons either in the absence of juglone, or in the presence of juglone (Fig 1E), suggesting that *flp-2* signaling promotes FLP-1 secretion by a mechanism that does not increase H2O2 levels in AIY. Consistent with this, intestinal overexpression of _flp-_2 had no effect on FLP-1::Venus secretion in the absence of juglone, but significantly enhanced the ability of juglone to increase FLP-1 secretion (Fig. 1D). We conclude that both elevated mitochondrial H2O2 levels and intact *flp-2* signaling from the intestine are necessary to increase FLP-1 secretion from AIY.”

(3) To confirm the activation of AIY neurons by FLP-2, measuring calcium activity in these neurons may be a robust approach. It would be beneficial to determine if synthetic FLP-2 can activate AIY neurons and subsequently induce an intestinal antioxidant response.

This is a great idea. We have begun to examine GCaMP fluorescence in AIY and we see responses to oxidative stressors. We think that this data is too preliminary at the moment to include here.

(4) The identification of the key receptors mediating the interaction between FLP-2 and AIY neurons, as well as the receptors in the gut that respond to FLP-1, would complete the signaling pathway and strengthen the study's conclusions.

We agree that this is an important question. Specifically, identifying the FLP-2 receptor and its site of action is a major priority. Since there are at least four different receptors that have been functionally or physically linked to FLP-2 and there are at least three FLP-2 peptides, unraveling the components acting directly downstream of FLP-2 will require further investigation that we feel is beyond the scope of this current study.

(5) Investigating whether direct manipulation of AIY neurons, through methods such as optogenetic activation or inhibition, can trigger the gut's antioxidant response would provide insight into the functional relevance of this neuronal activity.

Also an excellent idea. We previously published that Channelrhodopsin activation specifically in AIY indeed increases FLP-1 secretion, but we have not yet examined its effects on antioxidant responses in the intestine. This may require a more sustained activation of AIY than Channelrhodopsin can provide.

(6) For the analysis of intestinal Pges-1::GFP fluorescence, specifying the region of interest would enhance the precision of the data and the reproducibility of the results.

We analyze fluorescence intensity of a 16-pixel diameter circle in the posterior intestine (as indicated by the asterisks) and we have added this to the methods, we edited this paragraph:

“or transcriptional reporter imaging, young adult animals with indicated genotype were transferred into a 1.5mL Eppendorf tube with M9 buffer, washed three times and incubated in M9 buffer or 60uM working solution of juglone for 1h in dark on rotating mixer before recovering on fresh NGM plates with OP50 for 3h in dark at 20°C. The posterior end of the intestine was imaged with the 60x objective and quantification for average fluorescence intensity of a 16-pixel diameter circle in the posterior intestine was calculated using Metamorph.”

(7) Assessing the potential for pharmacological modulation of FLP-2 or H2O2 levels could provide valuable insights into therapeutic strategies aimed at enhancing the oxidative stress response.

Agreed.

(8) For improved clarity, it is suggested that the schematic currently presented in Figure S1A be integrated into Figure 2C, as this would facilitate the reader's comprehension of the experimental design and findings.

Moved.